**Projecting circum-Arctic excess ground ice melt with a sub-grid representation in the Community**
**Land Model**
Lei Cai[1], Hanna Lee[1], Kjetil Schanke Aas[2], Sebastian Westermann[2]
[1]NORCE Norwegian Research Centre, Bjerknes Centre for Climate Research, 5008, Bergen, Norway
[2]Department of Geosciences, University of Oslo, Oslo, 0315, Norway
*Correspondence to:* Lei Cai (leca@norceresearch.no)
**Abstract** To address the longstanding underrepresentation of the influences of highly variable ground
ice content on the trajectory of permafrost conditions simulated in Earth System Models under a warming
climate, we implement a sub-grid representation of excess ground ice within permafrost soils using the
latest version of the Community Land Model (CLM5). Based on the original CLM5 tiling hierarchy, we
duplicate the natural vegetated landunit by building extra tiles for up to three cryostratigraphies with
different amounts of excess ice for each grid cell. For the same total amount of excess ice, introducing
sub-grid variability in excess ice contents leads to different excess ice melting rates at the grid level. In
addition, there are impacts on permafrost thermal properties and local hydrology with sub-grid
representation. We evaluate this new development with single-point simulations at the Lena river delta,
Siberia, where three sub-regions with distinctively different excess ice conditions are observed. A triple-
landunit case accounting for this spatial variability conforms well to previous model studies for the Lena
river delta and displays a markedly different dynamics of future excess ice thaw compared to a single-
landunit case initialized with average excess ice contents. For global simulations, we prescribed a tiling
scheme combined with our sub-grid representation to the global permafrost region using presently
available circum-Arctic ground ice data. The sub-grid scale excess ice produces significant melting of
excess ice under a warming climate and enhances the representation of sub-grid variability of surface
subsidence on a global scale. Our model development makes it possible to portray more details on the
permafrost degradation trajectory depending on the sub-grid soil thermal regime and excess ice melting,
which also shows a strong indication that accounting for excess ice is a prerequisite of a reasonable
projection of permafrost thaw. The modeled permafrost degradation with sub-grid excess ice follows the
pathway that continuous permafrost transforms into discontinuous permafrost before it disappears,
including surface subsidence and talik formation, which are highly permafrost-relevant landscape
changes excluded from most land models. Our development of sub-grid representation of excess ice
demonstrates a way forward to improve the realism of excess ice melt in global land models, but further
developments require substantially improved global observational datasets on both the horizontal and
vertical distributions of excess ground ice.
**1.   Introduction**
Permafrost soils are often characterized by different types of ground ice that can exceed the pore
space (Brown et al. 1997; Zhang et al., 1999). The presence of such "excess" ground ice can alter the

permafrost thermal regime and landscape structure. Widespread thawing of permafrost is expected in a warmer future climate and modeling studies suggest large-scale degradation of near-surface permafrost at the end of the 21st century (Lawrence et al., 2008 & 2011). Melting of ground ice due to active layer thickening releases water in the form of surface runoff, subsurface flow, or both, causing surface subsidence and modifying the local hydrological cycle (West and Plug, 2008; Grosse et al., 2011; Kokelj et al., 2013; Westermann et al., 2016). In addition to containing ground ice, some permafrost soils store massive amounts of carbon, which could be released to the atmosphere in the form of greenhouse gases upon thawing (Walter et al., 2006; Zimov et al., 2006; Schuur et al., 2008), possibly making a positive feedback to amplify future climate change (Koven et al., 2011; Schaefer et al., 2014; Burke et al., 2013). The existence of excess ice and its distribution in permafrost can significantly affect the rate of permafrost thawing (Westermann et al., 2016; Nitzbon et al., 2020), and in turn, the rate of soil carbon release (Hugelius et al., 2014; Schuur et al., 2015; Turetsky et al., 2019). Therefore, better projections of excess ice melt are critical to improve our understanding of the impacts of permafrost thaw on corresponding climatic impacts.

Previous studies address excess ice modeling on the local or regional scale, in which the small study area makes it possible for detailed configurations of the cryostratigraphy of permafrost and excess ice based on observations. Simulations for the Lena river delta have retrieved the permafrost thermal dynamics fairly close to the observations with excess ice incorporated in the modeling (Westermann et al., 2016). A two-tile approach allowing lateral heat exchange between two land elements demonstrated that maintaining thermokarst ponds requires the heat loss from water to the surrounding land (Langer et al., 2016). A similar tiling approach has been applied to projecting the landscape changes due to permafrost thaw for ice-wedge polygons and peat plateaus with different features of ice melting and surface subsidence (Aas et al., 2019; Nitzbon et al., 2019).

On the global scale, the land components of Earth System Models (ESMs) have significant capabilities of representing key permafrost physics. In the Community Land Model (CLM), for example, the representation of permafrost-associated processes has been continuously improved. By including key thermal and hydrological processes of permafrost, the CLM version 4 (CLM4) has reasonably reproduced the global distribution of permafrost (Lawrence et al., 2008; Lawrence et al., 2012; Slater and Lawrence, 2013). Projections based on the CLM4 under its highest warming scenario (RCP8.5) have shown over 50% degradation of near-surface permafrost by 2100 (Lawrence et al., 2012). Moreover, the recently released CLM5 has more advanced representations of many biogeophysical and biogeochemical processes (Lawrence et al., 2019). A refined soil profile and upgraded snow accumulation and densification scheme in the CLM5 could contribute to simulating more realistic permafrost thermal regimes, whereas upgrades on biogeochemistry improve simulations of soil carbon release in response to permafrost thaw. In addition, an excess ice physics scheme has been implemented in CLM4.5 (CLM4.5_EXICE) by Lee et al. (2014), which allowed for the first-order simulation of surface subsidence globally by modeling excess ice melt under a warming climate.

The homogeneous distribution of excess ice throughout the grid cell in CLM4.5_EXICE (Lee et al.,
2014) could cause biases in thaw trajectories in the warming climate. In nature, excess ice forms in a
highly localized manner due to a variety of accumulation processes. For instance, segregated ice formed
during frost heave differs substantially in excess ice morphology from ice wedges that are formed from
repeated frost cracking and freezing of penetrating water.  Field measurements illustrate that the depth
distribution of ground ice can vary substantially on the order to 10-50 metres horizontally and 0-10 metres
vertically (Pascale et al., 2008; Fritz et al., 2011). The horizontal grid spacing of ESMs, on the other hand,
usually ranges from one to two degrees (~100-200 km horizontal scale), which makes it impossible to
represent localized excess ice. The mismatch in spatial scale between model and the real world raises
concerns for the reliability of excess ice modeling in ESMs. Aside from the homogenously initialized
excess ice in the grid cell, CLM4.5_EXICE initializes excess ice in the same soil depths globally (below
1m), regardless of the varying active layer thickness in circum-Arctic permafrost areas (Lee et al., 2014).
Such deficiencies in excess ice parameterization hamper global projections of permafrost thaw including
excess ice with ESMs.
To narrow the gap between the high spatial variability of excess ice and the coarse grid spacing in
the ESMs, we applied a sub-grid approach in representing excess ice in permafrost soils within the CLM5
to investigate how presence and melting of excess ice affect land surface physics under a warming climate.
We conducted idealized single-point simulations to examine the robustness of model development, and
furthermore conducted global simulations using a first-order estimate for the spatial distribution of excess
ice and associated cryostratigraphies. Due to the lack of information in global excess ice conditions, it is
not the aim of this study to accurately project excess ice melt and surface subsidence in the 21st century,
but rather to develop and present a functionable process within a land surface model  that can eventually
bring permafrost thaw modeling towards a higher degree of accuracy on a global scale. The CLM5 with
sub-grid excess ice representation developed through this study would be ready to serve as a proper
simulation tool on further advancing global excess ice modeling once new datasets become available.
**2.    Methodology**
**2.1 Sub-grid representation of excess ice in the CLM5**
The CLM5 model utilizes a three-level tiling hierarchy to represent sub-grid heterogeneity of
landscapes, which are (from top to bottom) landunits, columns, and patches (Lawrence et al., 2019).
There is only one column (the natural soil column) that is under the natural vegetated landunit, which
represents soil including permafrost. In this study, we modify the CLM5 tiling hierarchy by duplicating
the natural vegetated landunit, making extra landunits for prescribing up to three different excess ice
conditions in permafrost (Figure 1). The original natural vegetated landunit is considered as "natural
vegetated with no excess ice" (hereafter no ice landunit), while we denote the additional landunits as
"natural vegetated with low content of excess ice" (hereafter the low ice landunit), "natural vegetated
with medium content of excess ice" (hereafter the mid ice landunit), and "natural vegetated with high
content of excess ice" (hereafter the high ice landunit). The sub-grid initial conditions of excess ice are

imported as part of the surface data, which includes the variables of volumetric excess ice contents, depths of the top and bottom soil layer of added excess ice, and the area weights of the four landunits.

We adopted the excess ice physics from CLM4.5_EXICE (Lee et al., 2014), including thermodynamic and hydrological processes. The added excess ice is evenly distributed within each soil layer. Whereas the original CLM5 model already represents the dynamics of pore ice, our representation of excess ice physics only addresses the ground ice bodies that exceed soil pore space. The volumetric excess ice content in this study is defined as the ratio of the volume of excess ice in a soil layer to the volume of the whole soil layer. For example, a 50% volumetric content of excess ice means the excess ice body occupies 50% volume of a soil layer, while the rest of soil (and pore ice) occupies the other 50% volume of the soil layer. If not otherwise notified, the parameter of volumetric ice content in this manuscript refers only to that of excess ice bodies. After adding excess ice, the soil layer thickness increases accordingly. Because ice density is considered constant, the increase of soil layer thickness is linearly proportional to the volumetric content of excess ice. For example, adding an excess ice body with a 50% volumetric excess ice content doubles the soil layer thickness of the corresponding soil layer. The revised algorithm for thermal conductivity and heat capacity of soil involves the effects of added excess ice, while the revised phase change energy equation allows excess ice to melt. The meltwater adds to soil liquid water in the same soil layer, and it can move to the above layer if the original layer is saturated. Such numerical implementation replicates how the melt excess ice eventually converts to runoff and discharges from the soil in case of well-drained conditions. As excess ice melts, soil layer thickness decreases, which corresponds to surface subsidence due to excess ice melt. In our model parameterization, excess ice only melts and does not re-form since the applied excess ice physics does not account for the different ice formation processes.

Aside from sub-grid tiles for excess ice, we acknowledge that the version upgrade from CLM4.5 to CLM5 as the base model modifies the results of excess ice melt compared to the results from Lee et al. (2014). By default, CLM5 represents soil with a 25-layer profile, for which the top 20 hydrologically active layers cover 8.5 metres of soil. There are additional 10 soil layers and it is 4.7 metres deeper compared to the default hydrologically active soil layer profile in CLM4.5, not to mention the substantially more complex biogeophysical processes (Lawrence et al., 2019). Therefore, we developed the sub-grid representation of excess ice within the framework of the latest version of CLM. The duplicated landunits prolong computation time by roughly 10% compared to the original CLM5. We are, therefore, confident that our model development is highly efficient in addressing the sub-grid excess ice and subsequent permafrost thaw.

We examined the sensitivity of sub-grid excess ice initialization by conducting idealized experiments (see supplemental material). For overall the same amount of excess ice in one grid cell located in the same depth, a higher volumetric excess ice content along with a smaller area weight results in a later start of excess ice melt and a smaller melting rate. The different melting features from different sub-grid distribution of excess ice then leads to different hydrological impacts to the permafrost soil. The idealized experiments in this way verify the necessity of involving sub-grid configuration of excess ice

to the CLM that is with a typical horizontal grid spacing of 1-2 degrees. More details are available in the
supplemental material.

**2.2 Single-point simulations for the Lena river delta, Siberia**

We conduct single-point simulations for the Lena River delta and compare the CLM5 model results
to reference simulations with the CryoGrid3 model for the same location (Westermann et al., 2016).
Abundant background information is available on the soil and ground ice dynamics from both
observation and modeling, making the Lena river delta a suitable location to further evaluate our model
development. The Lena river delta can be broadly categorized into three different geomorphological units
that have distinctively different subsurface cryostratigraphies of excess ice (Schneider et al., 2009; Ulrich
et al., 2009). In the eastern and central part of the river delta, ground ice has been accumulated in the
comparatively warm Holocene climate. The subsurface sediments (hereafter denoted as "Holocene
ground ice terrain") are generally super-saturated with wedge ice that can extend up to 9 metres
underground with the volumetric contents of total ground ice (pore ice + excess ice) ranging from 60-
80% (Schwamborn et al., 2002; Langer et al., 2013). On the other hand, higher excess ice contents are
found in Pleistocene sediments in the Lena River Delta (hereafter the "Yedoma Ice complex"), which
are characterized by Yedoma type ground ice (Schirrmeister et al., 2013), which can reach depths of up
to 20-25 metres deep and volumetric contents of total ground ice as high as 90% (Schwamborn et al.,
2002; Schirrmeister et al., 2003 and 2011). Finally, the Northwestern part of the delta features sandy
sediments and is characterized by low excess ice contents (hereafter denoted the "no excess ice terrain";
Rachold and Grigoriev, 1999; Schwamborn et al., 2002).
We determine the area weights of excess ice landunits in one single point based on the spatial pattern
of three subregions (Fedorova et al., 2015). The cryostratigraphy and the volumetric contents of excess
ice strictly follow those in Westermann et al. (2016). Noting that the excess ice initialization scenario in
Westermann et al. (2016) does not necessarily represent the realistic excess ice condition for the Lena
river delta, the purpose of applying the same excess ice cryostratigraphy as in Westermman et al. (2016)
is to evaluate our model development by addressing intercomparisons between model results. Meanwhile,
we did not customize soil properties for different landunits as in Westermann et al. (2016), as our model
development does not support varying soil properties for different sub-grid landunits. We also directly
apply the snow accumulation physics in the CLM rather than customizing the snow density. By default,
the current model does not form thermokarst lakes as the meltwater from excess ice melt becomes surface
runoff and is removed from the grid cell. To apply the sub-grid representation, we initialize the case with
three landunits (the triple-landunit case) that respectively represent the three terraces in the Lena river
delta. We also initialize an "average ice single-landunit" case without the sub-grid representation of
excess ice. The excess ice amount for each soil layer in the average ice single-landunit case is initially
the same as that in the triple-landunit case. The volumetric content of excess ice is determined by spatial
averaging those for three excess ice landunits in the triple-landunit case. Detailed information on the
applied excess ice conditions for both cases is listed in Table 1.
We employed the single-point forcing data from in Westermann et al. (2016) for the Lena river delta
from 1901 to 2100, which is based on the CRU-NCEP (http://dods.extra.cea.fr/data/p529viov/cruncep/)
data set for the historical period (1901-2005) and the CCSM4 model output under the RCP4.5 scenario
for the projected period (2006-2100), but downscaled with in-situ observations. We run 100-year spin-
up simulations in order to stabilize the permafrost thermal regime after adding excess ice. Spin-up
simulations are produced by running the model with cycled 1901-1920 climatological data. The purpose
of spin-up simulations is to stabilize ground temperatures and volumes of excess ice bodies. The 100-
year length for spin-up is sufficient, as the model is run in Satellite Phenology (SP) mode that does not
involve slowly evolving biogeochemical processes such as soil carbon accumulation. Moreover, we
address idealized single-point simulations for additional permafrost locations with both continental and
maritime climate that showcase the difference to Lee et al. (2014), the results of which are included in
the Supplementary material.
**2.3 Global simulations of excess ice melt**
The information available for the spatial distribution of excess ice and associated cryostratigraphies
on the global scale is generally not as detailed as in the Lena river delta due to the lack of observations.
For our global simulations we employ the widely used "Circum-Arctic Map of Permafrost and Ground-
Ice Conditions" (hereafter the CAPS data; Brown et al., 2002) as data source, while we translate the
ground ice condition in the CAPS data to different excess ice stratigraphies as model input data. The
CAPS permafrost map categorizes the global permafrost area into classes coded by three factors (i)
permafrost extent (c = continuous, d = discontinuous, s = sporadic, and i = isolated), (ii) visible ground
ice content (h = high, m = medium, and l = low), and (iii) terrain and overburden (f = lowlands, highlands,
and intra- and intermontane depressions characterized by thick overburden cover, and r = mountains,
highlands ridges, and plateaus characterized by thin overburden cover and exposed bedrock), resulting
in more than 20 different varieties in permafrost characteristics (Figure 2). For the simulations, we only
use the CAPS distinction between the three classes: high, medium and low ice contents. We qualitatively
categorize excess ice types with typical cryostratigraphies for which observations are available,
recognizing that this is a crude first-guess of the global distribution of ground ice which needs to be be
improved in future studies.
The high ice CAPS classes (e.g. chf, chr, and dhf) in central and eastern Siberia, as well as in Alaska,
partly coincide with Yedoma regions (Kanevskiy et al., 2011; Grosse et al., 2013). The cryostratigraphy
of the high ice landunit is therefore broadly oriented at the excess ice contents and distribution in intact
Yedoma, which is characterized by massive ice wedges leading to typical average volumetric content of
total ground ice in the range from 60% to 90% (Schwamborn et al., 2002; Kanevskiy et al., 2011). We
therefore set the volumetric content of excess ice in the high ice landunit to 70%, and we put excess ice
in all the soil layers between 0.2 metres below the active layer and the bottom of hydrologically active
soil layer (8.5 metres). The onset depth of the excess ice just below the active layer is based on the
assumption of active ice aggradation which occurs at or below the permafrost table, e.g. the formation of
wedge or segregation ice. Initializing high excess ice content throughout the whole soil layer imitates the

cryostratigraphy of Yedoma type ice, while roughly 65% of the high ice landunit is located out of the observed Yedoma regions (Schuur et al., 2015). The effects, limitations, and potential improvements of this initialization scenario will be mentioned in the discussion section. For the low ice landunit, we assume both a significantly lower volumetric excess ice content and a smaller vertical extent of the excess ice body. The volumetric excess ice content is set to 25%, and we add excess ice at soil layers within 0.2 to 1.2 metres below the active layer, which in particular represents sediments with segregated ice (e.g. Cable et al., 2018), but also accounts for a wide range of different excess ice conditions found throughout the permafrost domain. For the mid ice landunit, we set the volumetric excess ice content to 45% and put excess ice within 0.2 to 2.2 metres below the active layer, making the volumetric excess ice content and vertical extent of which in between those for the low and high ice landunits. The cryostratigraphies determine that excess ice melt in the low ice landunit can result in a maximum of 0.36 metres of surface subsidence, while excess ice melt in the medium ice landunit can result in a maximum of 1.78 m of surface subsidence. For the high ice landunit, the surface subsidence can be more than 10 metres if all excess ice melts, which is expected to vary in space because of the different active layer thickness. For all three landunits, the active layer thickness is determined by the soil temperature profile by the end of the spinup in a no ice case, which is the simulation by the original CLM5 model without excess ice incorporated. Non-permafrost regions in the CAPS data are assigned the no ice landunit for 100% of their area. We emphasize that the prescribed cryostratigraphies are a first-order approximation that can by no means represent the wide variety of true ground ice conditions found in the permafrost domain. Nevertheless, this makes it possible to gauge the effect of excess ice melt on future projections of the permafrost thermal regime, when compared to "traditional" reference simulations without excess ice.

We design a tiling scheme prescribing the assignment of landunits for each CAPS class based on previous observations and empirical estimates (Table 2). All CAPS classes in this study are categorized into three levels of volumetric ice content (5%, 15%, and 25%) that are converted from the ranges (<10%, 10-20%, and >20%) in the original CAPS data. The goal of our tiling scheme is to determine a combination of area weights of three excess ice landunits for each CAPS class, making the spatially averaged volumetric content of excess ice the same as that for the CAPS class. We assume that all CAPS classes have the same area fraction (20%) of the low ice landunit, and the CAPS classes with a higher ice content are due to the existence of the landunits with a higher content excess ice. We make this assumption based on previous studies that the segregated ice is widely distributed in permafrost. Observational studies have found segregated ice bodies in various continuous permafrost regions across the circum-arctic including West Central Alaska (Kanevskiy et al., 2014), Nunavik, Canada (Calmels and Allard, 2008), and Svalbard (Cable et al., 2018). In discontinuous permafrost regions, segregated ice bodies also commonly exist underneath Palsas and Lithasas, including Fennoscandia (Seppälä, 2011), Altai and Sayan, Russia (Iwanhana et al., 2012), Himalayas (Wünnemann et al., 2008), and Mongolia (Sharkhuu et al., 1999). The volumetric content of visible segregated ice bodies mentioned above ranges widely from 10-50% (Gilbert et al., 2016).

Given the tiling scheme prescribed above, all CAPS classes are assigned a 20% area of low ice landunit. Correspondingly, the CAPS classes with 15% volumetric ice content are assigned another 14%

area weight for mid ice landunit on top of the CAPS classes with 5% volumetric ice content, while the CAPS classes with 25% volumetric ice are assigned another 22% area for high ice landunit on top of the CAPS classes with 15% volumetric ice content. The classes of "chf" and "chr" are the exceptions as their corresponding regions are typically with the landscape of Yedoma or ice wedge polygonal tundra or both (Kanevskiy et al., 2011; Gross et al., 2013). We therefore assign only the low and high ice landunits for these two CAPS classes. Summing up the landunit fractions for all the CAPS grid cells within each CLM grid cell obtains the area weights on the grid level that are stored in the surface data file. Figure 3 shows a schematic plot for the initialization scenario and the area covered by different excess ice landunits as the result of sub-grid excess ice initialization in the global simulation case. Note that excess ice for some regions (e.g. Southern Norway and the Alps) can completely melt out during the spinup period since the CLM initial condition prescribes overly warm (non-permafrost) soil temperature for these regions.

In this study, we define the grid cells or landunits with permafrost as the ones having at least one hydrologically active soil layer that has been frozen in the last consecutive 24 months. In this case, we define fully degraded permafrost when all landunits in one grid cell have an active layer thickness of more than 6.5 metres, recognizing that in reality permafrost at many localities may continue to exist at greater depths.. We also prepare a "grid-average ice case" by applying the same total amount of excess ice as in the sub-grid ice case in each soil layer, but using only one landunit instead of three that account for the sub-grid variability of excess ice. The volumetric content of excess ice in the single landunit is calculated as the spatial average of those in the three landunits in the triple-landunit case. This grid-average ice case provides a reference to evaluate the effects of the sub-grid excess ice representation on the global scale. Finally, we simulate a reference case without excess ice, denoted the "no ice case" in the following. Details on the three cases for the global simulations are listed in Table 3. All global cases are forced by the 3rd version of Global Soil Wetness Project forcing data (GSWP3; Kim et al., 2012), running in the Satellite Phenology (SP) mode. The International Land Atmosphere Model Benchmarking (ILAMB; Collier et al., 2018) project has indicated the superior performance of GSWP3 data forcing the CLM5 in the SP-only mode (http://webext.cgd.ucar.edu/I20TR/_build_090817_CLM50SPONLY_CRUNCEP_GSWP3_WFDEI/index.html). We conducted a 100-year spin-up using the 1901-1920 climatology before conducting historical period simulations covering 1901-2005. The anomaly forcing under the RCP8.5 scenario on top of the 1982-2005 climatology forces simulations in the projected period.

## 3. Result

### 3.1 Excess ice melt simulations for Lena River delta cryostratigraphies

By the end of the spinup in the triple-landunit case, the active layer thickness is 0.85 m, 0.55 m, and 0.45 m for the ice-poor terrain, the Holocene ice wedge terrain, and the Yedoma ice complex, respectively. On the other hand, the active layer thickness for the average ice single-landunit case is 0.85 m, which is the same as in the no excess ice terrain in the triple-landunit case. For the average ice single-landunit case, a small amount of excess ice (24kg m$^{-2}$) melts during the spinup period, resulting in 2.6 cm surface subsidence throughout the grid.

For the Yedoma ice complex, very little excess ice melt in the 1950s, and it stabilizes afterwards
until the late 2000s when substantial ice melt and surface subsidence starts to occur. For the Holocene
ground ice terrain, there is no excess ice melt before the late 2010s. By the year 2100, the Yedoma ice
complex has exhibited nearly 4 metres of surface subsidence, while the Holocene ground ice terrain has
about 0.6 metres of surface subsidence (Figure 4). For the average ice single-landunit case, the noticeable
excess ice melt and surface subsidence starts in the late 2010s, which creates about 0.5 metres of surface
subsidence by 2100. The magnitude of surface subsidence in the average ice single-landunit case is lower
than both the Holocene ground ice terrain and the Yedoma ice complex in the triple-landunit case.
On the grid scale, the total excess ice melt is higher in the average ice single-landunit case than in
the triple-landunit case (Figure 5). By the year 2100, the average ice single-landunit case has about 30
kg m$^{-2}$ more excess ice melt than the triple-landunit case. The difference in excess ice on the grid level
results from the different volumetric content of excess ice caused by the spatial averaging. In this way,
the sub-grid representation of excess ice can potentially also provide more detailed and realistic
representation of model variables on the grid level. This is particularly important for the CLM5, which
serves as the land component in Earth System Models, which requires the coupling between interacting
components on the grid level.
Compared to Westermann et al. (2016), the CLM5 with sub-grid excess ice simulates slightly less
(~ 20% less) surface subsidence by 2100 for both the central delta and ice complex. We consider this a
good agreement as we do not expect a closer fit of the model results due to substantial differences in the
model physics (for example, the Cryogrid3 simulations in Westermann et al. (2106) lack a representation
of the subsurface water cycle). What is in common between these two studies is the earlier start of excess
ice melt and more surface subsidence in the ice complex than in the central delta. The CLM5 with sub-
grid excess ice also exhibits the varying active layer thickness with different excess ice conditions as
Cryogrid3 does. These results suggest that the new model development enables small-scale variability in
excess ice melt and subsequent impacts in agreement with previously published modeling efforts.
**3.2 Global projection of permafrost thaw and excess ice melt**
Single-point simulations have shown that the varying excess ice cryostratigraphies for different
landunits result in sub-grid variabilities of excess ice melt and surface subsidence under the warming
climate. The same features remain in the sub-grid ice case within the global simulations that excess ice
in the low ice landunit can completely melt out throughout the circum-Arctic permafrost region by the
end of the 21$^{st}$ century (Figure 6). The modeled magnitude of surface subsidence is similar to the ~10 cm
surface subsidence observed in Barrow and West Dock in the early 21st century (Shiklomanov et al.,
2013; Streleskiy et al., 2017). The magnitude of surface subsidence is also comparable to the 1-4 cm
decade$^{-1}$ surface subsidence rate on average over the North Slope of Alaska observed by satellite
measurements since the 1990s (Liu et al., 2010). In comparison, the absence of surface subsidence for
Arctic Alaska modeled by Lee et al. (2014) is due to an overly deep (1 m deep) excess ice initialization
depth. By the year 2100, most ice in the medium ice landunit melts away in the sub-arctic region, while
there is less ice melt in the colder regions such as the North Slope of Alaska and the central Siberia. The

high ice landunit has the greatest surface subsidence among the three because of its high excess ice content, leading to 2-5 metres of surface subsidence by the year 2100.

The existence of excess ice modulates the thermal regime of permafrost soil and is a major control on permafrost degradation trajectories in a warming climate. Permafrost with excess ice consistently exhibits delayed permafrost degradation compared to the no ice case (Figure 7). For the no ice case modeled by the original CLM5, more than half of the permafrost area undergoes degradation by the end of the 21$^{st}$ century. By 2100, the only areas where permafrost remains are the North Slope of Alaska, Northern Canada, and the majority of the land area in Northern Siberia. The areas with remaining permafrost in the year 2100 under the RCP8.5 scenarios are substantially larger compared to the CLM4 simulations, in which nearly all permafrost in Eurasia becomes degraded (Lawrence et al., 2012). For the grid-average ice case, the presence of excess ice stabilizes the permafrost thermal regime and thus sustains a larger permafrost area on a global scale in the simulation. For example, permafrost areas in some subarctic regions in the eastern and western Siberia, as well as part of the Arctic coastal regions in Yukon Territory, Canada, remain in the grid-average ice case by 2100. Compared to the grid-average ice case, even more permafrost areas are sustained in the sub-grid ice case, most of which are located in southern Siberia. In the subarctic regions in Alaska and Northwest Canada as well as part of the central Siberia, permafrost degradation is delayed from the 2040s in the grid ice case to the 2080s in the sub-grid ice case. We emphasize that permafrost is only sustained according to the accepted temperature-based definition (ground material at temperature below zero for two consecutive years), but excess ice continuously melts in this process, which energetically is a different mode of permafrost degradation, similar to a negative mass balance of glaciers and ice sheets.

In the sub-grid ice case, the landunits with high excess ice contents lead to more grid cells for which permafrost conditions remain in the year 2100 compared to the grid-average ice case. On the other hand, permafrost with excess ice only covers a fraction of a grid cell. Among the permafrost degradation trajectories in the three global simulation cases (Figure 8), the sub-grid ice case can provide a more detailed picture on the timing of permafrost degradation. Grid cells become 'partially degraded permafrost' if landunits with excess ice still contain permafrost, which phenomenologically is a more realistic representation that also makes it possible to represent the permafrost distribution in the discontinuous and sporadic permafrost zones. On the other hand, only "fully degraded permafrost" and "remaining permafrost" can be distinguished for the no ice and grid-average ice case. Under the warming climate in the 21$^{st}$ century, the existence of excess ice, especially the high content of excess ice, has a stabilizing effect on soil temperature that delay the disappearance of permafrost on the sub-grid level. Therefore, by the year 2100, there are regions with partially degraded permafrost in between intact and degraded permafrost (Figure 8). For example, in western Siberia, the Pacific coastal area of eastern Siberia, Northwestern Canada, and along the Brooks Range in Alaska, taliks form for landunits with low excess ice contents which leads to partially degraded permafrost regions. Therefore, permafrost degradation exhibits a gradual transition from continuous to discontinuous permafrost, and to non-permafrost regions. Some of these regions also encounter substantial surface subsidence in the high ice landunit (> 5 m) (Figure 6).

We further compare the total permafrost area (defined as landunits with active layer thickness < 6.5
metres) in the three cases throughout time. The differences in permafrost area increase from the grid-
average ice case and sub-grid ice case to the no ice case at a rate of 1000 km$^2$ per year until 2050 (Figure
9). After 2050, the area difference of permafrost in the grid-average ice case and no ice cases rapidly
increases, which reaches nearly one million km$^2$ by 2100. In the sub-grid ice case, the rate of increase
remains relatively unchanged after 2050, resulting in an about 0.2 million km$^2$ larger permafrost area
than that in the no ice case.
**4. Discussion**
The aim of the sub-grid excess ice representation in the CLM5 is to facilitate long-term global
projection of excess ice melt and surface subsidence in the permafrost regions. Results from idealized
sensitivity experiments (see supplemental material) implies that overly low volumetric content of excess
ice, such as the grid-average ice case in this study and that in Lee et al. (2014), result in overly early start
of excess ice melt and an overly high melting rate. It is because the higher content of excess ice covering
a smaller area takes longer to absorb enough latent heat of fusion from the atmosphere before it can start
melting. A good model performance in this way relies not only on the updated sub-grid representation of
excess ice in the global land model, but also on retrieving accurate initial conditions of excess ice.
However,  the corresponding observational data for both background excess ice conditions and model
evaluation is sparse, considering especially that drastic excess ice melt as modeled until 2100 is only
observed in few locations today (e.g. Günther et al., 2015). In the following, we discuss the challenges
and limitations of the sub-grid excess ice framework, and how this sub-grid representation can potentially
help the development of other CLM components. Both single-point and global test simulations in this
study have shown that excess ice melts under a warming climate is sensitive to its initialization depth.
The active-layer-dependent excess ice initialization in this study in the global simulation (sub-grid excess
ice case) yields excess ice melt and surface subsidence rates in the early 2000s that are comparable to
observations. The lower depths of the assumed excess ice body control the termination of excess ice melt
which at the same time determines the onset of talik formation in many permafrost areas. Due to the
scarcity of observational data, it is unclear to what extent the cryostratigraphies assumed in our tiling
scheme can reproduce the true vertical extent of excess ice bodies at least in a statistical sense. Even so,
we manage to make the prescribed excess ice condition as close to the previous results as possible. Firstly,
our tiling scheme on the large scale strictly follows the CAPS data (Brown et al., 2002) in terms of the
volumetric excess ice content. Furthermore, statistics by Zhang et al. (2000) suggest the ranges of the
vertical extent of ice-rich permafrost of 0-2 metres and 2-4 metres respectively for the CAPS classes with
low (5%) and medium (15%) ice content. Comparatively, the vertical extents permafrost with excess ice
prescribed by our tiling scheme are respectively 1.36 metres and 3.78 metres for the same CAPS classes,
both of which lie within the ranges in Zhang et al. (2000). The vertical extent of ice-rich permafrost for
the high ice landunit is much higher than that (4-6 metres) in Zhang et al. (2000), but the unmelted part
of the ice bodies does not strongly affect the overall rate of excess ice melt, although the remaining ice
can slightly change soil temperature and moisture of the surrounding permafrost. We therefore imply
that our high ice landunit initialization would not induce a strong bias in excess ice melt projection in the
21st century.
Due to the lack of excess ice datasets and observational evidence, our projections of excess ice melt
and surface subsidence likely have biases that arise from the need to make empirical estimates and
simplifications for the excess ice initialization scenarios in the global simulation cases. For example, as
the CAPS data is mostly based on visible ice bodies (i.e., not pore ice) (Heginbottom et al., 1995), we
used the reported volumetric ground ice content in the CAPS data to approximate the volumetric content
of excess ice during model initialization.further the determination of volumetric contents of excess ice
for three landunits also results from sparse observations and empirical estimates. The prescribed excess
ice cryostratigraphies ignore ice morphology and the variation of volumetric content of excess ice with
soil depth, regarding excess ice as homogeneous within each assigned sub-grid ice content type (low,
mid, or high) (Figure 3, upper panel).. For the high ice landunit, we simplify the cryostratigraphy
initialization to Yedoma type ice, which prescribes overly thick excess ice bodies out of the Yedoma
regions (Schurr et al., 2015). A deficiency in the current version of source code prevents us from
initializing non-Yedoma wedged ice for the high ice landunit where it occurs outside of the Yedoma
region. Future versions of our model development will have more freedom in the stratigraphic
configuration of excess ice, which will make it possible to prescribe different cryostratigraphies of the
same landunit (e.g. the high ice landunit) for different locations. Because of the above shortcomings in
the excess ice initialization, we do not expect the modeled excess ice melt in this study to be an adequate
representation of reality. However, direct ingestion of new or improved observational data sets of excess
ice contents and cyostratigraphies would likely yield more accurate results. However, a spatially
distributed global dataset with quantitative information on excess ice stratigraphies does not exist at
present. We emphasize that for a better projection of excess ice melt, more observational data of excess
ice distribution and surface subsidence is required to further evaluate and validate the new model
implementation of excess ice. On the regional scale, Jorgenson et al. (2008) presented a permafrost map
of total ground ice volume for the uppermost 5 metres of permafrost based on both observations and
estimates for Alaska. In addition, O'Neill et al. (2019) compiled permafrost maps for Northern Canada
by paleographic modeling, mapping the abundances of three types of excess ice respectively. Further
improvements of model results depend on additional observationally constrained datasets of excess ice
conditions on the global scale.
The area weights of the excess ice landunits (Table 2) in the global simulation are obtained from the
higher-resolution CAPS points located within a CLM grid cell. However, complex landscape
development, such as thermokarst ponds, requires knowledge of the metre-scale distribution, for example
the extent and geometry of individual ice wedges (Langer et al., 2016; Nitzbon et al., 2019), which cannot
be represented with the still coarse-scale excess ice classes from the CAPS map. One possible solution
to represent this could be to include another layer of sub-grid tiles below the CLM landunit level, where
the individual tiles can interact laterally. This would allow for the representation of small-scale
permafrost features within a large-scale landunit with a given excess ice content. An example of how this
could work is given by Aas et al. (2019) who simulated both polygonal tundra and peat plateaus with a

two-tile interactive setup. This is also similar to the recent representation of hillslope hydrology by Swenson et al. (2019), where sub-grid tiles (on the column level in CLM) were used to represent different elements in a representative hillslope. In the future development of CLM, this could be part of a more generic tiling system where lateral heat and mass fluxes could be switched on and off to represent a wide range of land surface processes that are currently ignored or parameterized in LSMs. Fisher and Koven (2020) have discussed the challenges and opportunities in such an adaptive and generic tiling system. We would also advocate for enhancing current tiling schemes in such a direction, which could substantially improve the realism in the representation of permafrost landscapes in LSMs. However, the success of such a tiling approach will rely heavily on the availability of adequate observational data, further highlighting the need for observational efforts and close collaboration between field scientists and modelers.

The more detailed simulation of permafrost degradation trajectory with a sub-grid representation of excess ice also builds more potential on better modeling the permafrost-carbon feedback with biogeochemistry activated (CLM5BGC). Excess ice stabilizes the permafrost thermal regime, therefore alter the rate of carbon releasing from the permafrost (Shuur et al., 2008). Improved projections of permafrost warming could also enhance modeling of vegetation type changes (e.g. shrub expansion) that determines the nitrogen uptake to the atmosphere (Loranty and Goetz, 2012). On the other hand, the possibility to simulate surface subsidence and excess ice meltwater formation also opens the possibility of a more accurate representation of wetland formation. The increase in the area of wetland and soil moisture have an impact of the balance of $CH_4$ and $CO_2$ releasing from the permafrost as more organic matter could decompose in an anaerobic pathway (Lawrence et al., 2015; Treat et al., 2015). Compared to the parameterized inundated area simulation in the CLM5 (Ekici et al., 2019), a process-based wetland physics scheme together with the sub-grid representation of excess ice in this study would substantially contribute to the biogeochemical modeling over the circum-arctic area.

**5.  Conclusion**

This study develops a sub-grid representation of excess ice in the CLM5 and examines the impacts of the existence and melting of excess ice in the sub-grid scale in a warming climate. Extra landunits duplicated from the natural vegetated landunit in the CLM sub-grid hierarchy make it possible to prescribe up to three different excess ice conditions in each grid cell with permafrost.

A test over the Lena river delta showcases that the sub-grid representation of excess ice can retrieve the sub-grid variability of annual thaw-freeze state and the excess ice melt and surface subsidence through time. On the other hand, initializing excess ice homogeneously throughout the grid cell produces a smaller stabilization effect of excess ice to the permafrost thermal regime and the local surface subsidence under a warming climate. With a tiling scheme ingesting a global data set of excess ice condition into the CLM surface data, our model development shows the capability of portraying more details on simulating permafrost degradation trajectories. As excess ice thermally stabilizes the permafrost on the sub-grid scale, permafrost degrades with a trajectory from continuous permafrost to discontinuous permafrost, and finally to a permafrost-free area. The modeled global pattern of permafrost

therefore exhibits regions of discontinuous permafrost as the transition zone between the continuous permafrost and degraded permafrost.

This study, for the first time, used an ESM to project excess ice melt and surface subsidence and permafrost degradation with sub-grid variability. The approach of duplicating tiles at the landunit level instead of the column level allows more freedom for further developments in this direction. Furthermore, the new CLM tiling hierarchy has much more potential than representing more accurate excess ice physics as examined in this study. The accuracies of predicted excess ice melt and surface subsidence trends are limited at present by the available global-scale dataset and studies on excess ground ice conditions, thus further advancement of the excess ice modeling will rely on new or improved observational studies or datasets of the excess ground ice conditions at the global scale. The model development in our study, therefore, lays the foundation for further advances focusing on excess ice modeling and other processes in the CLM framework that could benefit from an improved sub-grid representation.

**Source code and data availability**

The original Community Land Model is available at https://github.com/ESCOMP/ctsm. The source code of model development in this study is available from the corresponding author upon request.

**Author contributions**

L.C conducted model development work and wrote the initial draft with additional contributions from all authors. H.L, S.W, and K.S.A provided ideas and help during the process of model development. H.L provided the code of excess ice physics in the earlier version of CLM. L.C prepared all figures.

**Acknowledgments**

This study is funded by the Research Council of Norway KLIMAFORSK program (PERMANOR; RCN#255331). K.S.A is supported by the Research Council of Norway EMERALD project (RCN#294948). We thank Sarah Chadburn for helpful comments and suggestions in preparing this manuscript.

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

**Table 1: The excess ice initialization scenario in each of the three terraces (landunits) for the Lena**
**River delta, as well as that for the single-landunit excess ice initialization case.**

| Depth (after adding ice) | Volumetric Ice content | Area weight |
|:---:|:---:|:---:|
| No excess ice terrain | | |
| N/A | 0% | 24.6% |
| Holocene ground ice terrain | | |
| 0.9-9 m | 65% | 66.6% |
| Yedoma ice complex | | |
| 0.6-20 m | 90% | 8.8% |
| Average ice single-landunit case | | |
| 0.6-0.9 m | 7.92% | 100% |
| 0.9-9 m | 51.21% | 100% |
| 9-20 m | 7.92% | 100% |



**Table 2: The tiling scheme prescribing area weights of landunits for each CAPS class. The detailed**
**CAPS classes are shown in Figure 2.**

| Overall visible ground ice content for each CAPS point | Tiling scheme (area weights for each excess ice category) | Eligible CAPS types |
|---|---|---|
| 5% | 80% no excess ice; 20% Low | clf; clf; slf; ilf; clr; dlr; slr; ilr |
| 15% | 58% no excess ice; 20% Low; 22% Medium | cmf; dmf; smf; imf; dhr; shr; ihr |
| 15% | 66% no excess ice; 20% Low; 14% High | chr |
| 25% | 44% no excess ice; 20% Low; 22% Medium; 14% High | dhf; shf; ihf |
| 25% | 52% no excess ice; 20% Low; 28% High | chf |

Note: For each class, the first letter is for the permafrost extent, the second for the excess ice content, and the third
for the terrain and overburden, following Brown et al. (2002).

**Table 3: List of simulations conducted for this study.**

| Cases | Description |
|---|---|
| *Single point cases for the Lena river delta* | |
| Triple-landunit case | Applying the sub-grid representation of excess ice. Three natural vegetated landunit initialized. |
| Average ice single-landunit case | Not applying the sub-grid representation of excess ice. Only one natural vegetated landunit initialized. The grid-mean excess ice content for each soil layer in the only landunit is calculated by spatially averaging those in different landunits in the triple-landunit case. |
| *Global simulation cases* | |
| No ice case | Not adding any excess ground ice (the original CLM5 simulation). |
| Sub-grid ice case | Applying the sub-grid representation of excess ice. A tiling scheme helps to "translate" excess ice conditions in the CAPS data to fit what the CLM5 requires. |
| Grid-average ice case | Not applying the sub-grid representation of excess ice. The grid-mean excess ice content for each soil layer is calculated by spatially averaging those in different landunits in the sub-grid ice case. |




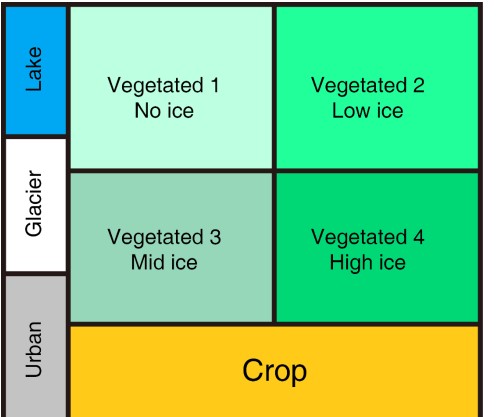

**Figure 1: Modification of the CLM5 tiling hierarchy on the landunit level containing four natural**
**vegetated landunits for different excess ice conditions.**

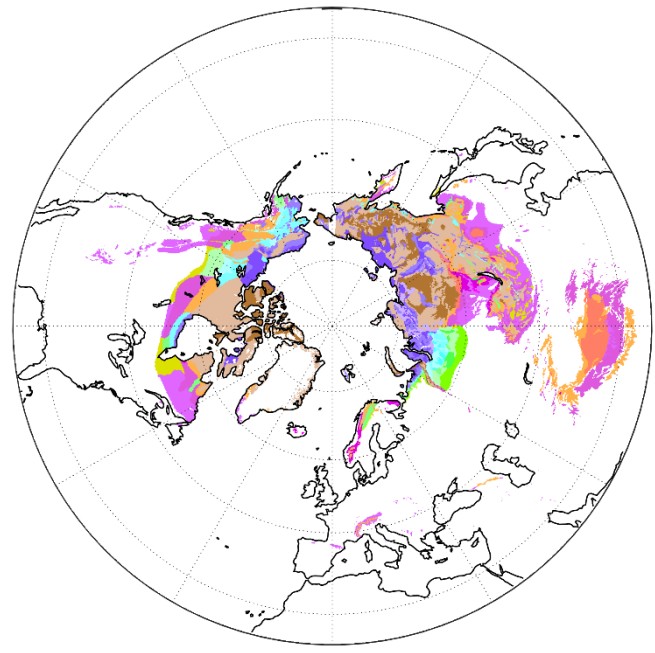

**Permafrost area classification**

| Permafrost Extent | Ground Ice Content (percent by volume) | | | | |
|---|---|---|---|---|---|
| | Lowlands, highlands, and intra-and intermontane depressions | | | Mountains, highlands, ridges, and plateaus | |
| | 25% | 15% | 5% | 15% | 5% |
| Continous (100%) | chf | cmf | clf | chr | clr |
| Discontinous (70%) | dhf | dmf | dlf | dhr | dlr |
| Sporadic (30%) | shf | smf | slf | shr | slr |
| Isolated (5%) | ihf | imf | ilf | ihr | ilr |

\* Letter code naming: The first letter is for the permafrost extent, second for the ground excess ice concent, and the thrid for the terrain and overburden.


**Figure 2: Spatial distribution of excess ground ice in the Northern Hemisphere modified from**
**Brown et al. (2002). Compared to the original data, permafrost extents and ground ice contents**
**are converted to definite numbers (percentages) for model computation.**


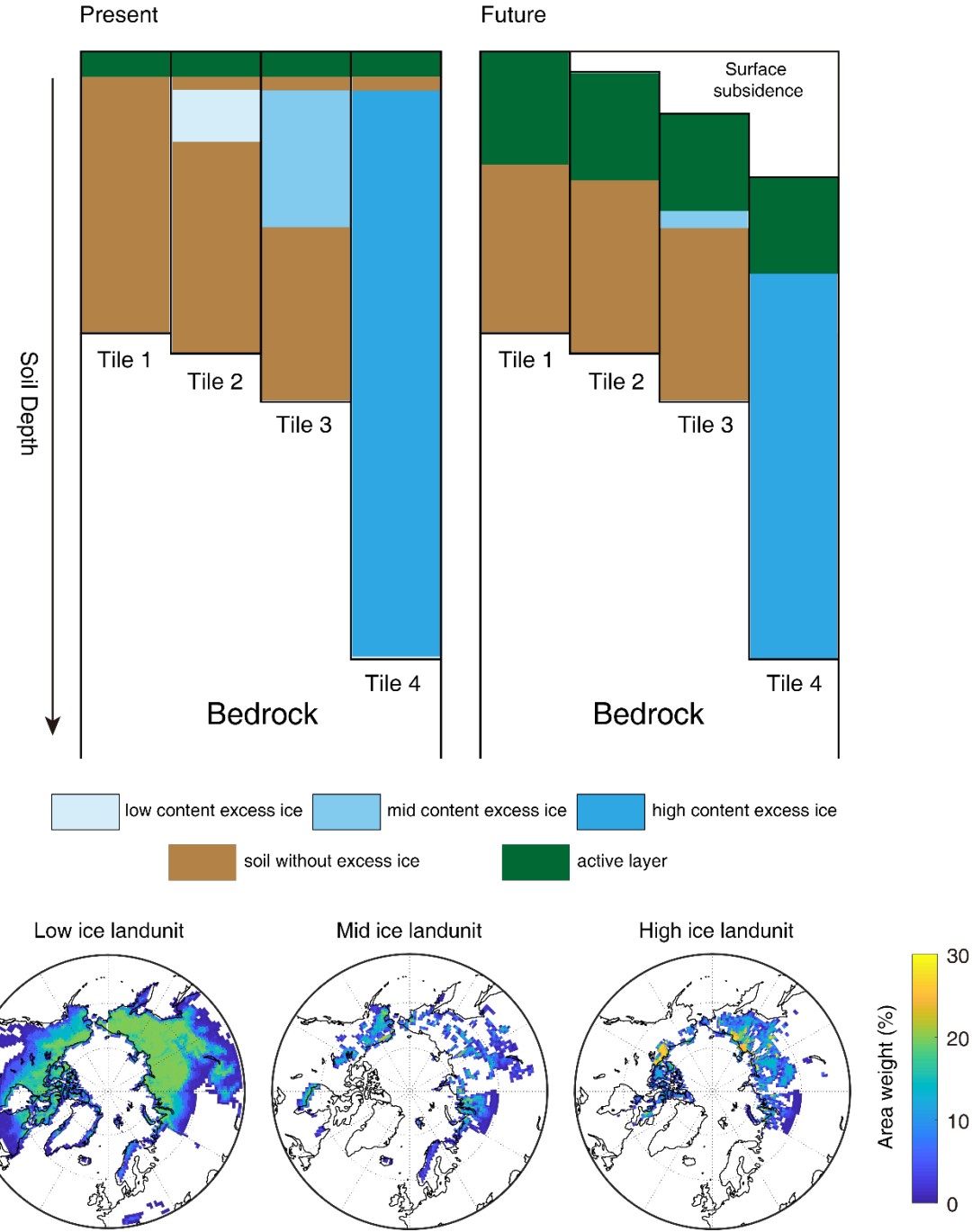

**Figure 3. Schematic representation of the sub-grid excess ice initialization scenario, and maps**
**showing the area weight (%) occupied by different excess ice landunits, i.e. the initial condition of**
**excess ice in the global simulation.**

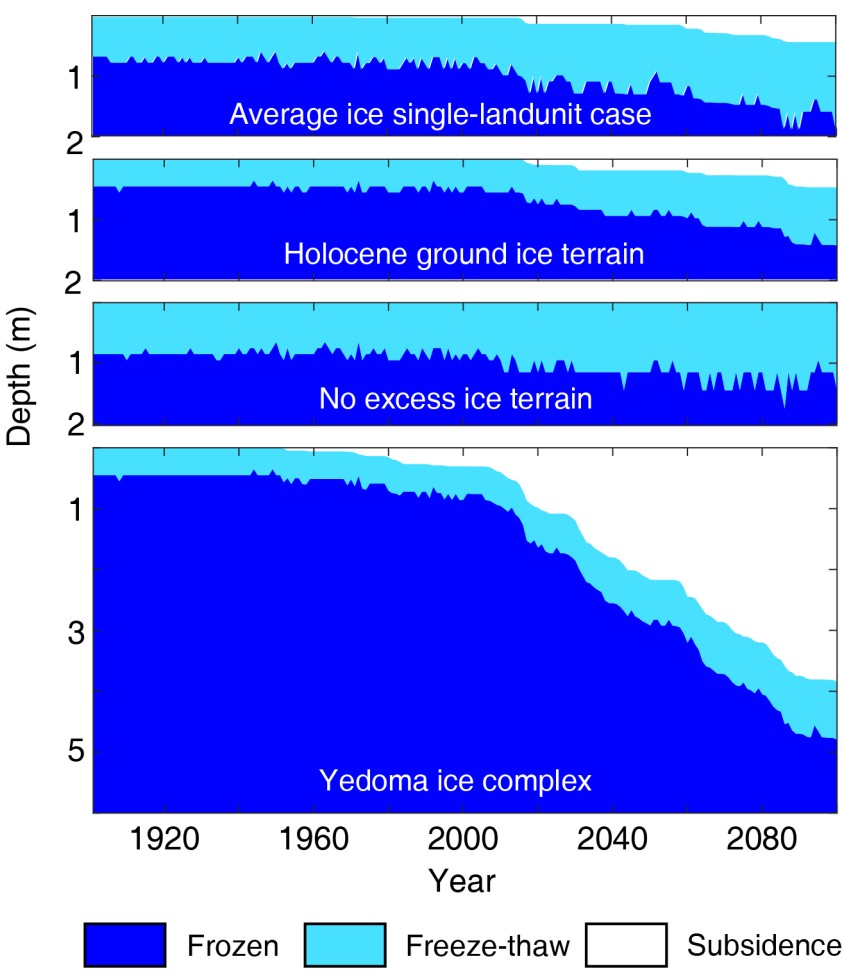


Figure 4. Annual freeze-thaw state for the three terraces for the triple-landunit case, as well as for

the average ice single-landunit case.


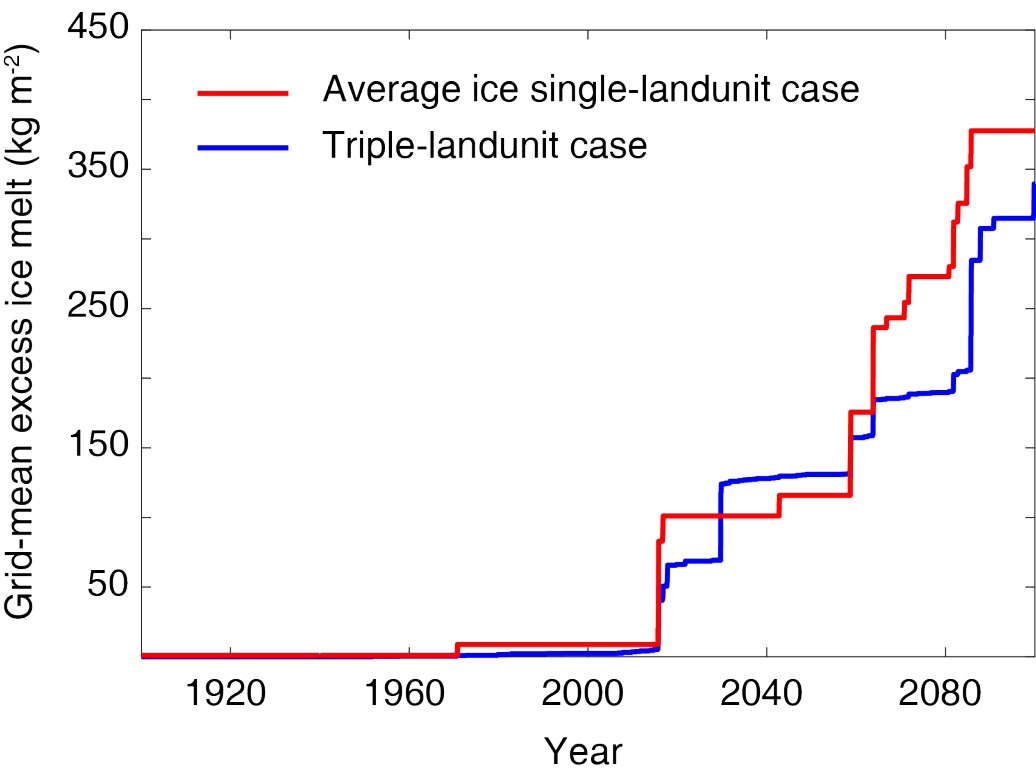


**Figure 5. Grid-mean excess ice melt since 1900 for the single-point cases over the Lena river delta**
**with and without the sub-grid excess ice initialization.**




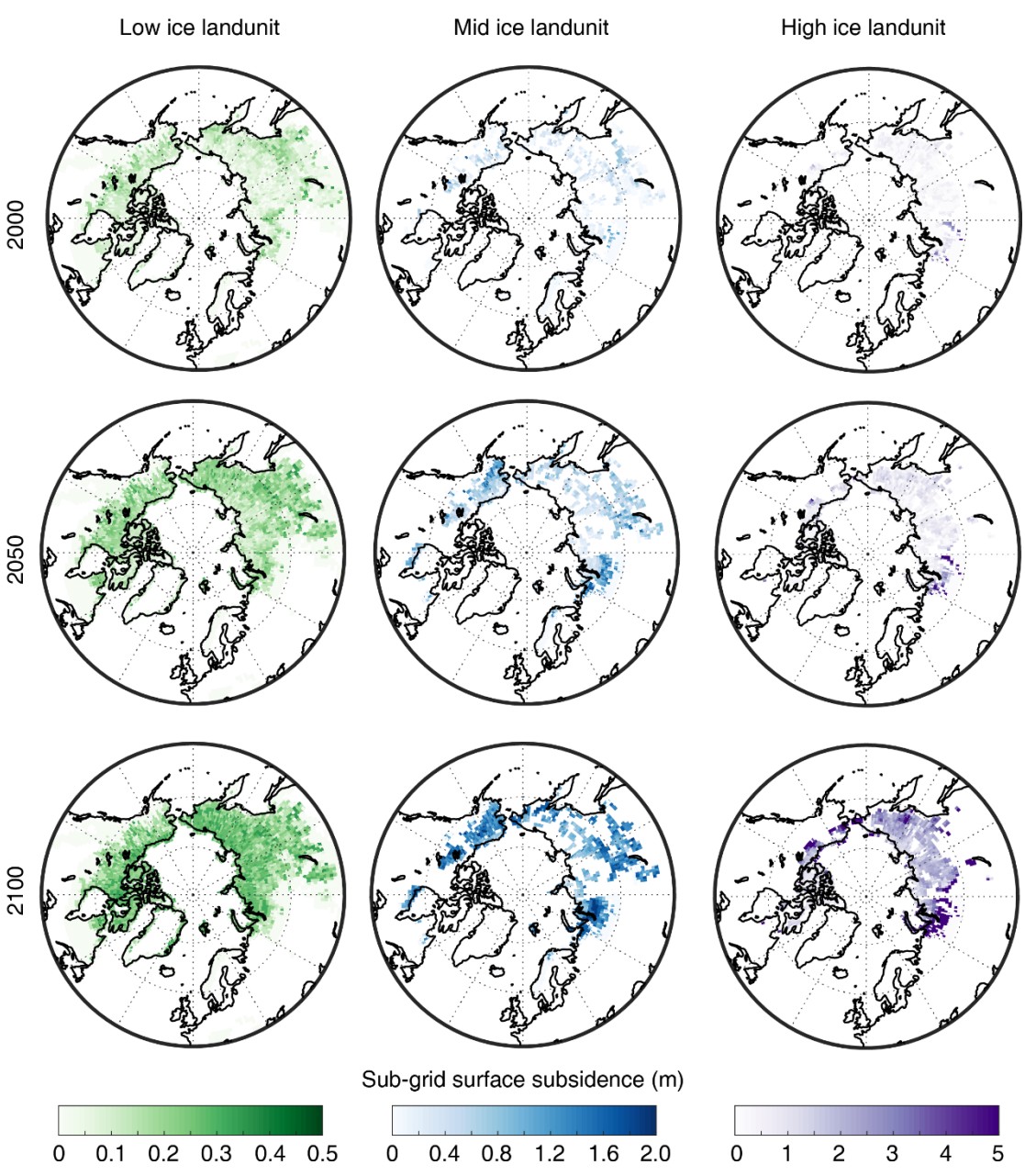


**Figure 6. Maps showing sub-grid surface subsidence (m) in 2000, 2050, 2100 in the low, mid, and**
**high excess ice landunits in the sub-grid ice case.**

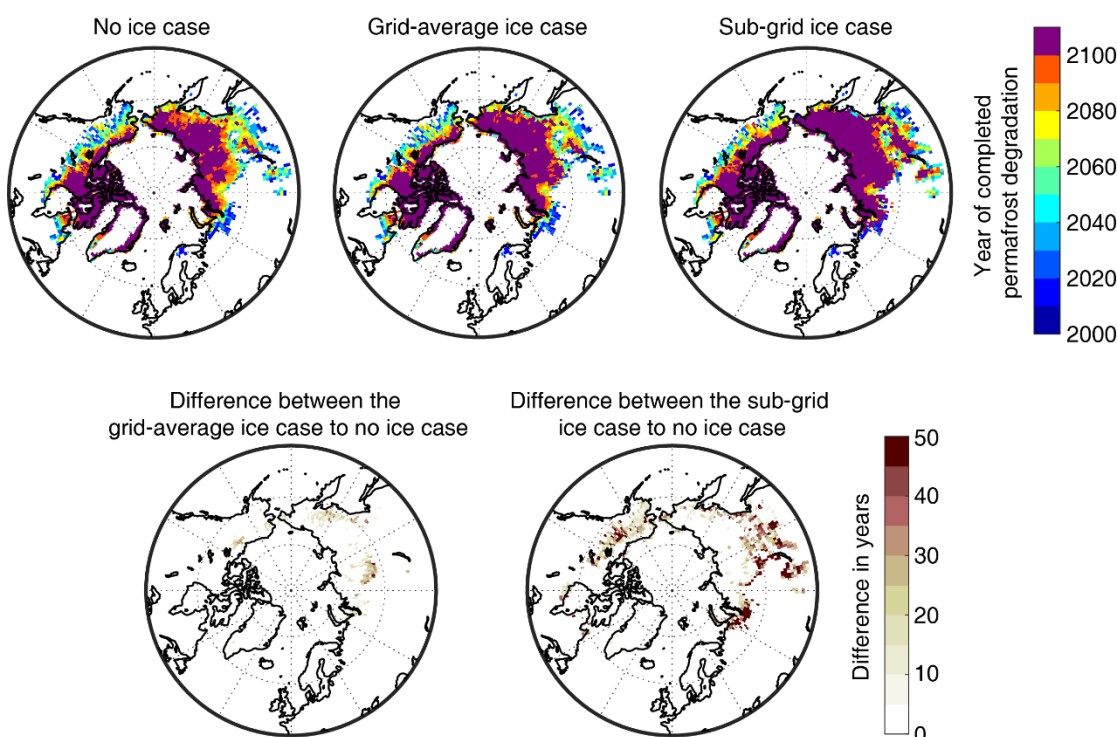


**Figure 7. Maps showing the year of completed permafrost degradation (upper set of three maps),**
**as well as the differences between cases (lower set of two maps). The purple color indicates the**
**existence of permafrost in these grid cells by 2100. The difference in years is provided only for grid**
**cell with completed permafrost degradation before 2100.**

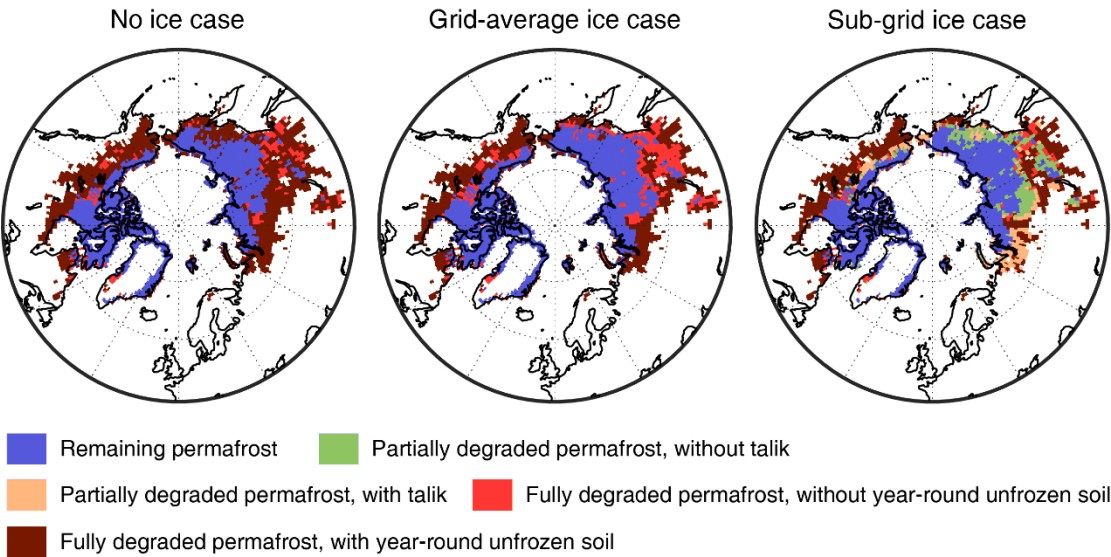

Remaining permafrost  Partially degraded permafrost, without talik

Partially degraded permafrost, with talik  Fully degraded permafrost, without year-round unfrozen soil

Fully degraded permafrost, with year-round unfrozen soil

**Figure 8. Maps of different stages of permafrost degradation diagnosed from the model output by the year 2100. "Year-round unfrozen soil"in the fully degraded permafrost region is defined as the part of degraded permafrost in which the soil temperature never decrease below 0 ℃ in any time of year, which is in the same manner as talik in the permafrost area.**

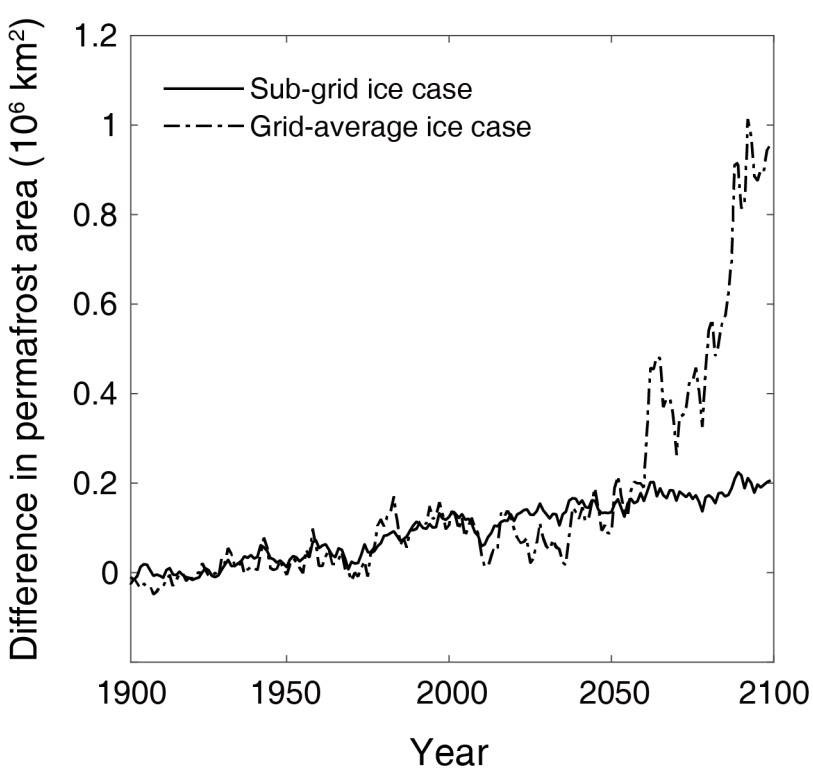


**Figure 9. Difference in modeled permafrost area versus time between the sub-grid ice case and no**
**ice case, as well as between the grid-average ice case and no ice case.**