# Peer review of "Projecting circum-Arctic excess ground ice melt with a sub-grid representation in the Community"

_The Cryosphere, 2020_

## Referee Comment (RC1) · Anonymous Referee #1 · 1 Jun 2020

This is a resubmission of a previous discussion paper that was retracted by the authors following review: https://www.the-cryosphere-discuss.net/tc-2019-230/. For context, my previous review is available here: https://doi.org/10.5194/tc-2019-230-RC1.

The single-point modelling has been changed to simulate 3 geomorphic units in the Lena River delta, rather than Yakutsk and the North Slope of Alaska in the initial submission. The global simulations include comparison of a no ice case, sub-grid representation case, and a grid-average case.

My main criticisms of the first submission were that (a) the results were not validated in any meaningful way, (b) the empirical basis for the parameterization of excess ice

was lacking, and (c) that there was not a clear comprehension of empirical ground ice studies and knowledge of ground ice conditions.

I have read up to the results section and made several observations pertaining to points (b) and (c) above. The points below do little to reassure me of my concerns with (b) and (c) from the previous version. Furthermore, in my previous review I pointed out that references mentioned in text were missing from the reference list. I expected such a simple item would be remedied, but in the first paragraph of the introduction alone, the following references are missing from the list: Walter et al. (2006); Schaefer et al. (2011).

Given these concerns, I have not formally reviewed the results or discussion.

1. It is unclear from the text whether the authors appreciate the difference between "excess ice content", "volumetric ice content", and "visible ice content", as the terms are seemingly used interchangeably or confused.

In different places in the paper, the authors have indicated the CAPS values represent volumetric ice content, excess ice content, and visible ice content. The authors have misinterpreted the legend for the Circum-Arctic Map of Permafrost (CAPS) in their Figure 2. They have altered the legend from the original map by removing the clause stating "visible ice in the upper...", and now only indicate "Ground Ice Content: percent by volume". They report ice contents from the Circum-Arctic Map as volumetric ice content (lines 216, 224) in the text. Then, in the figure 2 caption, they suggest the CAPS values represent the "Spatial distribution of excess ground ice" – very confusing. The CAPS legend, and the Permafrost Map of Canada (Heginbottom et al. 1995) legend on which the CAPS compilation is based, both clearly indicate that the ice content reported is the visible ice content (as the authors correctly indicate on line 177). The legend on the Heginbottom et al. (1995) map indicates this visible ice percentage accounts for "segregated ice, intrusive ice, reticulate ice veins...". The percentages on the maps do not correspond to volumetric ice content (in the strict sense), which also

include the pore ice fraction.

Lines 185 to 190, the authors report that Yedoma is "characterized by massive ice wedges leading to typical average volumetric ice contents in the range from 60% to 90%" (line 188). They then state: "We therefore set the volumetric excess ice content to 70%". Nowhere in the text do the authors mention the soil porosity, which is key to estimating excess ice content given only volumetric ice content. For example, if one assumes a soil porosity of 0.5, then volumetric ice contents of 60-90% represent excess ice contents of about 10-40%. Assuming an excess ice content of 70% based on volumetric ice contents of 60-90%, as presented above, is problematic. I refer the authors to Harris et al. (1988) for definitions of volumetric ice content and excess ice content.

Other examples that seemingly use the terms interchangeably: Line 137-138 "volumetric ice contents ranging from 60-80%" and in the next sentence, "higher excess ice contents are found in Pleistocene sediments. . ."; Line 193 "For the low ice landunit, we assume both a significantly lower volumetric ice content and a smaller vertical extent of the excess ice body"; Table 1. The caption reads "excess ice initialization scenario", but the table header indicates "Volumetric Ice content". Presumably, porosity is available, so why not also present the readers with excess ice content?

Finally, the term "ice content" (line 198) is also used on its own, as is "Overall Ground ice content" in Table 2, further complicating interpretation by the reader. What type of ice content? I'm left wondering throughout.

2. The authors suggest that high ice classes mapped on the Circum-Arctic Map of Permafrost and Ground ice Conditions (CAPS), designated in the submitted paper text as chf, chr, and dhf partly coincide with Yedoma areas and are "broadly oriented at the excess ice contents and distribution in intact Yedoma" (line 186-87).

The high ice landunit is considered representative of Yedoma. I'd like to point out the two maps below. Figure 1 shows the areas of chf, chr, dhf highlighted in red. Figure 2

shows the distribution of Yedoma from Schuur et al (2015). The area mapped as chf, chr, and dhf is much more extensive than areas mapped as Yedoma. For example, a large portion of the Canadian Arctic Archipelago (CAA) is mapped as chr: continuous permafrost that has high visible ice content (>10%) and thin overburden cover (5-10m) and exposed bedrock. Most of the CAA was glaciated and includes no Yedoma. It therefore seems inappropriate to me that vast areas such as this include a considerable fraction of the high ice landunit in the modelling that represents Yedoma.

The high ice landunit cryostratigraphy (70% excess ice in the upper ~8 m), may reasonably represent ice-rich Pleistocene deposits where permafrost has aggraded syngenetically, or local areas where large bodies of buried glacial ice occur just below the permafrost table. However, I can't think of situations where 70% excess ice content in the upper 8-10m would be reasonable for other deposits in which permafrost has formed epigenetically, given the typical decline in ice content with depth in epigenetic permafrost (e.g., French and Shur, 2010; Fig.2; Gilbert et al, 2019). I realize the authors acknowledge that the cryostratigraphies prescribed in the simulations are a coarse first-order approximation. However, the assumption that areas mapped with high ice content on the CAPS include significant areas where ground ice content is similar to thick Yedoma deposits, including those defined on the CAPS map as chr, seems particularly unrealistic and poorly justified.

It would have been simple to overlay CAPS and Yedoma areas in a GIS and examine the overlap within chf, chr, and dhf to better inform and substantiate landunit parameterizations/area weights.

3. The authors provide a rationale for the excess ice content in the high ice landunit (for global simulations), which is commented on above, but provide little rationale for the medium and low ice content landunits (lines 193-200). One reference to an empirical study is provided (Line 197). The authors indicate that the excess ice content and distribution for the low ice landunit "account for a wide range of different excess ice conditions found throughout the permafrost domain" (line 197-198). It would have benefitted the reader if some of these excess ice conditions were elucidated, with pertinent references.

4. The authors state that subsidence of "more than 10 meters" (line 203) could occur if all ice melted from the high ice landunit in the global simulations. Earlier, the authors indicate that "we put excess ice in all the soil layers between 0.2 meters below the active layer and the bottom of hydrologically-active soil layer (8.5 meters)". As it is written, >10 m of subsidence is implied from thaw of <8.5m of ground.

5. The authors indicate that abundant field data in the Lena River delta provide a good basis for initializing ice conditions in refocused single-point simulations. I fully agree that simulations in areas with good available data is crucial. However, the authors in fact report no measurements of excess ice content anywhere in section 2.2 (only some volumetric ice contents are provided). It would benefit the reader to have some of these examples if there is abundant field data.

I am also confused by the authors' interpretation of the data that is presented in this section. For example, in Line 136 the authors indicate that ice wedges extend to 9 m depth in the Holocene terrain unit, and that there are volumetric ice contents of 60-80%, citing Schwarmborn et al. (2002) and Langer et al. (2013). Schwarmborn et al. (2002) indicate much smaller ice wedges in the Holocene sediments: "and subaerial or buried ice wedges of 2–3m in height and width are common." (p. 123), and I cannot find wedge dimensions in Langer et al. (2013). I can only find mention of ice wedges that extend deeper (5-10 m) in the Ice Complex (Yedoma) unit in Schwarmborn et al. (2002).

The volumetric ice contents of 60-80% reported for the Holocene unit are seemingly from Langer et al. (2013, p.13) who indicate: "The elevated rims are usually covered with a dry moss layer underlain by wet sandy peat soils featuring massive ice wedges. The volumetric water/ice content of the peat soils typically ranges from 60 to 80%.". This value appears to refer to the volumetric ice content of the mineral soil

between ice wedges, rather than to an average representative value for a terrain unit or cross-section that includes both the icy soil matrix and ice wedges. At the scale of the modelling, this is what is pertinent, otherwise the contribution to ice content in the upper permafrost from ice wedges is not accounted for.

6. Line 106: "The added ice is evenly distributed within each soil layer". In Figure 3, ice it not depicted as evenly distributed in the cross-sectional diagrams. Tile 4 shows large ice wedges, tile 3 a discontinuous (across the landunit) body of ice. The model does not represent ice in this way. These diagrams should reflect that ice is evenly distributed and consistent with the depictions showing "Present" and "Future" conditions.

References French, H. and Shur, Y., 2010. The principles of cryostratigraphy. Earth-Science Reviews, 101(3-4), pp.190-206.

Gilbert, G.L., O'Neill, H.B., Nemec, W., Thiel, C., Christiansen, H.H. and Buylaert, J.P., 2018. Late Quaternary sedimentation and permafrost development in a Svalbard fjord‐valley, Norwegian high Arctic. Sedimentology, 65(7), pp.2531-2558.

Harris, S.A., French, H.M., Heginbottom, J.A., Johnston, G.H., Ladanyi, B., Sego, D.C. and Van Everdingen, R.O., 1988. Glossary of permafrost and related ground-ice terms.

Heginbottom, J.A., Dubreuil, M.A. and Harker, P.A., 1995. Canada, Permafrost. National Atlas of Canada. Natural Resources Canada, 5th Edition, MCR, 4177.

Langer, M., Westermann, S., Heikenfeld, M., Dorn, W. and Boike, J., 2013. Satellite-based modeling of permafrost temperatures in a tundra lowland landscape. Remote Sensing of Environment, 135, pp.12-24.

Schuur, E.A., McGuire, A.D., Schädel, C., Grosse, G., Harden, J.W., Hayes, D.J., Hugelius, G., Koven, C.D., Kuhry, P., Lawrence, D.M. and Natali, S.M., 2015. Climate change and the permafrost carbon feedback. Nature, 520(7546), pp.171-179.

Schwamborn, G., Rachold, V. and Grigoriev, M.N., 2002. Late Quaternary sedimentation history of the Lena Delta. Quaternary international, 89(1), pp.119-134.

[Figure]

[Figure]

**Fig. 1.** Mapped by reviewer from the CAPS GIS files.

b

Major river deltas
Yedoma largely unaffected by thaw cycles
Region of potential yedoma distribution
Thick sediments
Continuous permafrost
Discontinuous permafrost

**Fig. 2.** From Schuur, E.A., McGuire, A.D., Schädel, C., Grosse, G., Harden, J.W., Hayes, D.J., Hugelius, G., Koven, C.D., Kuhry, P., Lawrence, D.M. and Natali, S.M., 2015. Climate change and the permafrost car

---

## Referee Comment (RC2) · Anonymous Referee #2 · 9 Jul 2020

Comments on "**Projecting circum-Arctic excess ground ice melt with a sub-grid representation in the Community Land Model**" by Lei Cai et al submitted to The Cryosphere.

**General**

Permafrost soils usually contain large amount of ground ice. Its melting has significant impacts on infrastructure, landscape and hydrology. Ground ice also affects the timing and speed of permafrost thaw. This paper modelled the effects of ground ice on permafrost thaw using a sub-grid representation in the Community Land Model. They first test the implementation in Lena River delta. It shows that using three land units of different ground ice provides more realistic results than using one average ice land unit. The modelled thawing depths also very different among the three land units and from using the average ice content. Then they implemented the representation across the circum- arctic region using four land units (no ice, low, mid and high ice) and compared with the results using average ice content. The results shows more realistic pathways of permafrost degradation and a different total area with permafrost comparing to using average ice. The circum-arctic excess ice data are rough, the CAPS dataset is a very broad generalization of the complex ground ice conditions and how to use the dataset is not straightforward. However, this study does show some progress to include ground ice in a more realist way than previous studies (no excess ice, or using average for an entire grid) and it provides a general range of the large-scale impacts of such sub-grid differences. The paper is well prepared in language and figures.

**Major points**

The test study shows very different active-layer thicknesses among the three land units and from the one-unit with average-ice (Figure 4). The paper did not provide much about the results of active-layer thickness for the circum-arctic modelling. It would be important to add this part in the results and analysis. Observations on ground subsidence is sparse and highly depend on the local conditions. An improved modelling of active-layer thickness would provide some support evidence about the usefulness of including excess ice in sub-grids.

"Compared to the grid average ice case, even more permafrost areas are sustained in the sub-grid ice case" (Line 313-314). However, Figure 9 shows the permafrost area difference between sub-grid case and no ice case is similar to the difference between the average ice case and no ice case before the 2050, after that the latter reached about 1 million km2. That means the permafrost areas under average ice case and sub-grid ice case are similar before the 2050s. After that, the modelled permafrost area under average ice case is larger than under sub-grid ice case. In the last two panels in Figure 7, the shaded area in the second panel seems larger than the second panel. That is not consistent with the results in Figure 9. Not sure whether my understanding is correct. Any way, it would be useful and interesting to provide more explanation and analysis about the differences among these three cases (no ice, average ice and sub grid ice).

The data about ground ice is rough and how to use the current data is based on some assumptions or artificial choices. It would important to indicate that uncertainties more clearly in the text (the paper already indicated that at different places).

**Minor points**

Line 28-29: delete "enhance" or "improve".

Lines 42-44, "The existence of excess ice and its distribution in permafrost can significantly affect the rate of permafrost thawing". It would be useful to add some references here.

Line 58: "over generations". It seems strange to say model versions as "generations". It would be clearer to say "in recent years" or so.

Line 67: "Separate from this", revised to "In addition"

Line 71-74. Check the grammar for this long sentence.

Line 74-95: "the depth distribution of ground ice can vary substantially on the order to 10-50 meters horizontally 75 and 10 meters vertically". Is the depth to the top of ground ice or also including the thickness of ground ice? Probably you want to say both. Check and consider revising the sentence.

Line 165: "Satellite Phenology (SP) mode", I do not know what is that. Some explanation would be helpful.

Line 220: "Have the same area fraction of low ice landunit", You may add "(20%)" to make it clearer. What is the reason behind this assumption?

You must have a percentage of land as no excess ice as the total percentage is less than 100% in Table 2 (e.g., for 5% CAPS, the no excess ice area would be 80%). If that is the case, it would be clearer to indicate the no excess ice areal percentage in Table 2, and the scheme actually uses four landunits (as shown in Figure 1) rather than three. For the grid-average ice case, you used the average of the three land units (Line 242) or the four land units?

Figure 3. The legend is in km2. You may provide the area of a grid or using % of the area of a grid.

Line 259-260: "A small amount of excess ice (24kg/m2) melts during the spinup period", which case?

Lines 302-303: "We define the permafrost degradation in this study as when all the landunits in one grid cell has an active layer thickness greater than 6.5 meters". That is different from the sentence in line 238. Probably the sentence in lines 302-303 is for how you treat the grid in figure 7. If so you can indicate its applications.

Line 350: "as projected until 2100", probably revise to "as we modelled". No observations beyond present.

Line425, 438: "modelling", "modelled", be consistent with "Modeling" and "Modeled"

---

## Author Response (AR1)

Dear Dr. Lei Cai et al.,

Thank you for your responses to the Author's Comments. I recommend that you submit a
revised manuscript according to address the referee's comments and few more that I have
made that stem from their comments.

Re: Referee #2

Referee #2 seems generally satisfied with the revised version of the manuscript, but offers
a few major points and several minor points that you have indicated will be addressed in
your next revision. Your replies seem to address the questions adequately, with the exception
of the reply to R2's comment about your assumption on Line 220:

"Line 220: "Have the same area fraction of low ice landunit", You may add "(20%)" to make
it clearer. What is the reason behind this assumption?"

Please make the reason behind this assumption clear in the revised version of the manuscript.
I do not fully understand the logic as written in your Author's Reply.

Authors' reply: We have elaborated on the assumption in the revised manuscript.

Re: Referee #1

Referee #1 was not that satisfied with the revised version of the manuscript, and did not
provide comments on the Results and Discussion as a consequence. R1's major criticisms
still stem from initial concerns raised in the first round of reviews that there was (i) not enough
demonstration of the empirical basis for the parameterization of excess ice, and (ii) that
knowledge and understanding gained from empirical ground ice studies needs to be made
clearer to the reader. R1 provided several detailed points to help crystalize the issues.

Having read through your Author's Reply to R1, I have the following suggestions for you
to incorporate into your next revision:

As you say you intend to, please ensure that you state early on a clear scope and what the
great limitations are.

Authors' reply: We have clarified the scope and limitations of this study (model development)
in both the introduction and discussion sections in the revised manuscript.

Regarding terminology around ground ice content, I hope that it is made clear in the revised
version. In the Author's Reply you say "we emphasize that volumetric ice content in this
study refers only to excess ice bodies", but this is not how most readers think of volumetric
ice content. Volumetric ice content is pore ice + excess ice. If you use the term "volumetric
excess ice", it must always mean the volume ice in excess of the pore ice. The terminology
must be reconciled, made clear to the reader, and agree with how the terms are commonly
defined.

Authors' reply: We have now refined the terminology throughout the manuscript. Now we
have changed the "volumetric ice content" to "volumetric content of excess ice" or
"volumetric excess ice content" in order to emphasize that the ice content we refer to is
only for the excess ice bodies that exceed the soil pore space. For the volumetric content of
ground ice that includes both excess ice and pore ice, we keep using the term "volumetric
content of total ground ice" or "total volumetric ice content" for clarification. One exception
is that the ice content of the CAPS data, for which we keep using "volumetric ice content"
to keep the terminology consistent with the source data. As suggested by the R1, in the revised manuscript we clarify that the "volumetric ice content" is approximately equal to
the "volumetric content of excess ice" in our study because the production of CAPS data is
mostly based on the visible excess ice bodies (Heginbottom et al. 1995).

Regarding excess ice content outside of the Yedoma Region, I suggest that you try to follow
R1's comment and develop a way to better initialize wedge ice types that are not within
Yedoma deposits. R1 makes a good suggestion to "overlay CAPS and Yedoma areas in a
GIS and examine the overlap within chf, chr, and dhf to better inform and substantiate
landunit parameterizations/area weights", which should be follow up on in the revised
manuscript. Perhaps also have a look at O'Neill et al. (The Cryosphere, 13, 753–773, 2019,
https://doi.org/10.5194/tc-13-753-2019).

Authors' reply: We agree that the reviewer has given constructive suggestions on the
initialization of high ice landunit, while we did not follow his advice for several reasons. One
reason we have mentioned in the reply to R1 that this new scenario on high ice landunit does
not decrease the uncertainty of excess ice initialization on the global scale. There is also a
technical issue for the current version of our model development that it does not support
freely configuring excess ice volumetric contents for landunits in different grid points, which
means that all the high ice landunits in the same domain have to have the same excess ice
cryostratigraphy. Making it possible to initialize different cryostratigraphies for the same
excess ice landunit but in different locations needs substantial changes in the source code and
surface data variables, which probably should be regarded as a new version. Since all model
development work needs several updates since the release of the first version, we have
planned to add the function of freely configuring excess ice stratigraphy for each grid
point/landunit in the upcoming version, which is also dependent on the new excess ice
datasets. We have added text in the discussion section about the limitations and potential
improvements of our current model development on excess ice initialization scenario and
excess ice landunits assignments.

In your response to R1's comment 3, you state a caveat about the availability of ground ice
information helpful to your sub-grid representation:

"As we mentioned, there is a lack of dataset on ground excess ice with enough information
helpful for our sub-grid excess ice representation. For this reason, this is our best effort to
make a possible scenario of excess ice distribution based on the best dataset (the CAPS data)
at this time, even though it only provides generalized information and has been released for
more than 20 years. Due to the lack of adequate information in excess ice distributions, the
purpose of this study is not to make an accurate estimate of excess ice melt and surface
subsidence in the 21st century, but rather to develop a functionable process within a land
surface model on a global scale. Once there is a new generation of excess ice dataset, the CLM
with sub-grid excess ice representation is able to be operational and give more accurate
projections of excess ice melt and surface subsidences."

Please make sure that you state something to this effect in the introduction. Stating the clear
purpose will set up clear expectations from the reader. This caveat should also be echoed in
the Discussion. Given all of the uncertainty, and the goal of making a functioning process
within the land surface model, it would be instructive to include a sensitivity analysis of the
effects of differing sub-grid excess ice representation.

Authors' reply: We have added stuff about the scope and limitations of this model
development study in both the introduction and discussion. We keep our idealized simulation and analysis of the sensitivity of different sub-grid distribution of excess ice (North Slope of
Alaska and Yakutsk) in the previous version of manuscript (tc-2019-230) and put it into the
supplemental material.

Regarding R1's point 4, I don't follow the calculation in your example. If the original soil
layer is 7.5 m thick (between 1 and 8.5 m), and you increase it's volume by 70%, 7.5 x 1.70
is 12.75. adding back the first 1 m of ground gives 13.75 m of hydrologically active soil, no?
Not sure how one arrives at 18.5 m of hydrologically active soil. In any case, please make
sure that the added content in the main text makes the model design clearer.

Authors' reply: In the example taken in the reply to R1, since the soil layer after adding excess
ice has the volumetric content of excess ice of 70%, the original soil layer takes 30% of the
volume. In this way, the new thickness of soil is calculated as $1+7.5\times0.7\div0.3=18.5$ (m).

Regarding R1's point 5, it perhaps stems from the initial set up of the reader's expectations.
You have indicated that you have added clarification in the new text to address this point. I
additionally suggest that if the purpose of the manuscript is "not to retrieve realistic excess
ice melt, but rather to compare the model results from this study and from Westermann et al.
(2016)", then this purpose needs to be stated explicitly, and the inclusion of comparisons to
empirical studies should be carefully done so as not to give the wrong impression.

Authors' reply: We have added clarification in this section that the single-point experiments
over the Lena River Delta are just for model evaluation so that we want to initialize excess
ice exactly the same as in Westermann et al. (2016) in order to compare the model results in
the two studies.

Regarding R1's point 6, I agree that the schematic should show how the model actually
represents ground ice in the grid point. Show the "squeezing". If the added ice is "evenly
distributed within each soil layer", please show this distribution. It is expected that this
representation is an abstraction, and not reality.

Authors' reply: After some discussion, we decided to remove the upper left panel of figure 3
to avoid misunderstandings. We think that the upper right part of figure 3 has already presented
the concept of "squeezing". We have also added text in the methodology section about
"squeezing" to make our statement clear.

Please note the references kindly provided by R1 and incorporate where appropriate.

Authors' reply: We have incorporated the references provided by R1.

I look forward to receiving your revised manuscript in the near future.

Best regards,

Peter

Anonymous Referee #1 This is a resubmission of a previous discussion paper that was retracted by the authors following review: https://www.the-cryosphere-discuss.net/tc-2019-230/. For context, my previous review is available here: https://doi.org/10.5194/tc-2019-230-RC1.

The single-point modelling has been changed to simulate 3 geomorphic units in the Lena River delta, rather than Yakutsk and the North Slope of Alaska in the initial submission. The global simulations include comparison of a no ice case, sub-grid representation case, and a grid-average case.

My main criticisms of the first submission were that (a) the results were not validated in any meaningful way, (b) the empirical basis for the parameterization of excess ice was lacking, and (c) that there was not a clear comprehension of empirical ground ice studies and knowledge of ground ice conditions.

I have read up to the results section and made several observations pertaining to points (b) and (c) above. The points below do little to reassure me of my concerns with (b) and (c) from the previous version. Furthermore, in my previous review I pointed out that references mentioned in text were missing from the reference list. I expected such a simple item would be remedied, but in the first paragraph of the introduction alone, the following references are missing from the list: Walter et al. (2006); Schaefer et al. (2011).

Given these concerns, I have not formally reviewed the results or discussion.

Authors' reply: We appreciate your valuable comments which have contributed much to this new revision of our manuscript. Here we respond to your two (remaining) main concerns. The individual points have been addressed below.

First of all, we have tried to clarify the scope of the study in the new manuscript, which is to provide a proof-of-concept for how heterogeneous excess ground ice can be represented in a global Land Surface Model (LSM) used in Earth System Models (ESMs). While much work remains before excess ice is represented in a fully satisfactory way in ESMs, we believe this study represents an important step forward compared to the current generation models, which for the most part fully ignores excess ground ice (only representing pore ice). Much development of CLM (and other LSMs) in recent years have aimed at mechanistic representation of key features, even when improvements to the model performance cannot be demonstrated. As an example, the latest version of CLM showed an apparent degradation in representation of snow water equivalent at global scale, despite mechanistic improvements in snow physics (Lawrence et al. 2019). We believe our model enhancement is in line with this aim, as it accounts for the effect of heterogeneous excess ice on hydrology and thermal properties in a physically sound way, even though there are great limitations in the current study, especially related to the initialization of excess ice.

Secondly, we have now clarified the terminology. As you correctly pointed out, the previous version of the manuscript was ambiguous here, which understandably gave concern about the use of observational studies. We want to highlight here again that we fully recognize the limitations in excess ice initiation in our study. The observational studies listed in the manuscript are not intended to be replicated here but are used to motivate the use of three broad excess ice classes, which should be revisited in future studies.

1. It is unclear from the text whether the authors appreciate the difference between "excess ice content", "volumetric ice content", and "visible ice content", as the terms are seemingly used interchangeably or confused.

In different places in the paper, the authors have indicated the CAPS values represent volumetric ice content, excess ice content, and visible ice content. The authors have misinterpreted the legend for the Circum-Arctic Map of Permafrost (CAPS) in their Figure 2. They have altered the legend from the original map by removing the clause stating "visible ice in the upper. . .", and now only indicate "Ground Ice Content: percent by volume". They report ice contents from the Circum-Arctic Map as volumetric ice content (lines 216, 224) in the text. Then, in the figure 2 caption, they suggest the CAPS values represent the "Spatial distribution of excess ground ice" – very confusing. The CAPS legend, and the Permafrost Map of Canada (Heginbottom et al. 1995) legend on which the CAPS compilation is based, both clearly indicate that the ice content reported is the visible ice content (as the authors correctly indicate on line 177). The legend on the Heginbottom et al. (1995) map indicates this visible ice percentage accounts for "segregated ice, intrusive ice, reticulate ice veins. . .". The percentages on the maps do not correspond to volumetric ice content (in the strict sense), which also include the pore ice fraction.

Lines 185 to 190, the authors report that Yedoma is "characterized by massive ice wedges leading to typical average volumetric ice contents in the range from 60% to 90%" (line 188). They then state: "We therefore set the volumetric excess ice content to 70%". Nowhere in the text do the authors mention the soil porosity, which is key to estimating excess ice content given only volumetric ice content. For example, if one assumes a soil porosity of 0.5, then volumetric ice contents of 60-90% represent excess ice contents of about 10-40%. Assuming an excess ice content of 70% based on volumetric ice contents of 60-90%, as presented above, is problematic. I refer the authors to Harris et al. (1988) for definitions of volumetric ice content and excess ice content.

Other examples that seemingly use the terms interchangeably: Line 137-138 "volumetric ice contents ranging from 60-80%" and in the next sentence, "higher excess ice contents are found in Pleistocene sediments. . ."; Line 193 "For the low ice landunit, we assume both a significantly lower volumetric ice content and a smaller vertical extent of the excess ice body"; Table 1. The caption reads "excess ice initialization scenario", but the table header indicates "Volumetric Ice content". Presumably, porosity is available, so why not also present the readers with excess ice content?

Finally, the term "ice content" (line 198) is also used on its own, as is "Overall Ground ice content" in Table 2, further complicating interpretation by the reader. What type of ice content? I'm left wondering throughout.

Authors' reply: We agree with the referee's comments that the terminology about ice content is somewhat unclear throughout the manuscript that could lead to misunderstanding of the main purpose of this study. But we do not believe we misrepresented the physical properties of ground ice overall when incorporating them into the structure of the large scale land surface model. The physical properties of ground ice used in our model development is only for the excess ice bodies that exceed the pore space of soil. In our model development, we do not address pore ice physics because it is already represented in the original CLM model, with the output variable named "soilice". The melting of "soilice" in the CLM5 does not cause surface subsidence as this ice only exists as part of pore space. Therefore, we emphasize that volumetric ice content in this study refers only to excess ice bodies. We agree that directly applying the groun ice content in the CAPS data is not necessarily an accurate way, while we have to make sufficiently simple classes of ice content levels to avoid over-parameterization. We think that using the volumetric ice content provided by the CAPS data is generally valid for the purpose of this research since the CAPS data is based mostly on "visible" ice bodies (Heginbottom et al. 1995). We have clarified the definition of "volumetric excess ice content" following Harris et al. (1989) in the methodology section. We have also discussed the limitation of applying the ice content values in the CAPS data in our model development in the discussion section.

2. The authors suggest that high ice classes mapped on the Circum-Arctic Map of Permafrost and Ground ice Conditions (CAPS), designated in the submitted paper text as chf, chr, and dhf partly coincide with Yedoma areas and are "broadly oriented at the excess ice contents and distribution in intact Yedoma" (line 186-87).

The high ice landunit is considered representative of Yedoma. I'd like to point out the two maps below. Figure 1 shows the areas of chf, chr, dhf highlighted in red. Figure 2 C3 shows the distribution of Yedoma from Schuur et al (2015). The area mapped as chf, chr, and dhf is much more extensive than areas mapped as Yedoma. For example, a large portion of the Canadian Arctic Archipelago (CAA) is mapped as chr: continuous permafrost that has high visible ice content (>10%) and thin overburden cover (5-10m) and exposed bedrock. Most of the CAA was glaciated and includes no Yedoma. It therefore seems inappropriate to me that vast areas such as this include a considerable fraction of the high ice landunit in the modelling that represents Yedoma. The high ice landunit cryostratigraphy (70% excess ice in the upper ~8 m), may reasonably represent ice-rich Pleistocene deposits where permafrost has aggraded syngenetically, or local areas where large bodies of buried glacial ice occur just below the permafrost table. However, I can't think of situations where 70% excess ice content in the upper 8-10m would be reasonable for other deposits in which permafrost has formed epigenetically, given the typical decline in ice content with depth in epigenetic permafrost (e.g., French and Shur, 2010; Fig.2; Gilbert et al, 2019). I realize the authors acknowledge that the cryostratigraphies prescribed in the simulations are a coarse first-order approximation. However, the assumption that areas mapped with high ice content on the CAPS include significant areas where ground ice content is similar to thick Yedoma deposits, including those defined on the CAPS map as chr, seems particularly unrealistic and poorly justified.

It would have been simple to overlay CAPS and Yedoma areas in a GIS and examine the overlap within chf, chr, and dhf to better inform and substantiate landunit parameterizations/area weights.

Authors' reply: We agree that overlaying the Yedoma coverage information and the CAPS data can give a better interpretation over the Yedoma region. However, the excess ice content and located depth of ice wedges out of the Yedoma region is still unclear and lacks observational support. Although we fully acknowledge the importance of accurately representing different Yedoma cover in the model, for the sake of model representation of permafrost thaw processes, having an accurate projection over the Yedoma region does not improve the projections of the excess ice melt over the whole circum-arctic in general. Since the main purpose of our study is to represent permafrost thaw processes on a global scale, we make a decision to initialize different kinds of ice wedged ice as "Yedome type ice". As we understand this may not be fully representing reality, we added discussion on how these initialization scenarios brings uncertainties to surface subsidence projections in the discussion section. The high excess ice content and the relatively cold climate where the high ice landunit is located make the wedged ice almost impossible to melt out completely by the end of the 21st century. The remaining part of the excess ice at the bottom has little effect on the surface subsidence. In this way, surface subsidence projections by 2100, initializing Yedoma type ice at the Yedoma region does not substantially affect the final result in our model simulations.

As we write in the discussion section, the purpose of simulation on top of this first-order scenario is to show how our model development can represent permafrost thaw processes on a global scale. Our modeling result shows that the current version of the CLM5 can represent permafrost degradation process with a wide range all the way from continuous to discontinuous permafrost and even no permafrost with the developed sub-grid representation of excess ice. The surface subsidence in the sub-grid representation produces greater heterogeneity to the land surface. Talik forming can also be retrieved during the degradation process. All of the above are novel progresses that no other state-of-the-art global land models can represent.

3. The authors provide a rationale for the excess ice content in the high ice landunit (for global simulations), which is commented on above, but provide little rationale for the medium and low ice content landunits (lines 193-200). One reference to an empirical study is provided (Line 197). The authors indicate that the excess ice content and distribution for the low ice landunit "account for a wide range of different excess ice conditions found throughout the permafrost domain" (line 197-198). It would have benefitted the reader if some of these excess ice conditions were elucidated, with pertinent references.

Authors' reply: The scenario we designed for the low ice landunit is based on previous studies that the segregated ice is widely distributed throughout the permafrost area. We have added more reference that segregated ice has been widely distributed throughout the permafrost area, both continuous and discontinuous permafrost (Line 239-246). We also provide an additional empirical excess ice volumetric content (25%) and located depth (ALT+0.2 ~ALT+1.2) to the low ice landunit. For the mid ice landunit, the volumetric content of excess ice and located depths are set in between the low and high ice landunits, which are also based on empirical data. As we mentioned, there is a lack of dataset on ground excess ice with enough information helpful for our sub-grid excess ice representation. For this reason, this is our best effort to make a possible scenario of excess ice distribution based on the best dataset (the CAPS data) at this time, even though it only provides generalized information and has been released for more than 20 years. Due to the lack of adequate information in excess ice distributions, the purpose of this study is not to make an accurate estimate of excess ice melt and surface subsidence in the 21st century, but rather to develop a functionable process within a land surface model on a global scale. Once there is a new generation of excess ice dataset, the CLM with sub-grid excess ice representation is able to be operational and give more accurate projections of excess ice melt and surface subsidences.

4. The authors state that subsidence of "more than 10 meters" (line 203) could occur if all ice melted from the high ice landunit in the global simulations. Earlier, the authors indicate that "we put excess ice in all the soil layers between 0.2 meters below the active layer and the bottom of hydrologically-active soil layer (8.5 meters)". As it is written, >10 m of subsidence is implied from thaw of <8.5m of ground.

Authors' reply: We have mentioned in the methodology section (Line 115) that the soil layer depth increases accordingly after adding excess ice. In this way, the soil thickness with excess ice added is thicker than 8.5 meters. For example, adding high ice landunit (70% volumetric excess ice content) in the soil layers with the original depths between 1 and 8.5 meters can make the thickness of hydrologically-active soil 18.5 meters in total. > 10 m of subsidence is therefore possible in the simulation. We have added the content above in the main text to make the model design clearer.

5. The authors indicate that abundant field data in the Lena River delta provide a good basis for initializing ice conditions in refocused single-point simulations. I fully agree that simulations in areas with good available data is crucial. However, the authors in fact report no measurements of excess ice content anywhere in section 2.2 (only some volumetric ice contents are provided). It would benefit the reader to have some of these examples if there is abundant field data.

I am also confused by the authors' interpretation of the data that is presented in this section. For example, in Line 136 the authors indicate that ice wedges extend to 9 m depth in the Holocene terrain unit, and that there are volumetric ice contents of 60- 80%, citing Schwarmborn et al. (2002) and Langer et al. (2013). Schwarmborn et al. (2002) indicate much smaller ice wedges in the Holocene sediments: "and subaerial or buried ice wedges of 2–3m in height and width are common." (p. 123), and I cannot find wedge dimensions in Langer et al. (2013). I can only find mention of ice wedges that extend deeper (5-10 m) in the Ice Complex (Yedoma) unit in Schwarmborn et al. (2002).

The volumetric ice contents of 60-80% reported for the Holocene unit are seemingly from Langer et al. (2013, p.13) who indicate: "The elevated rims are usually covered with a dry moss layer underlain by wet sandy peat soils featuring massive ice wedges. The volumetric water/ice content of the peat soils typically ranges from 60 to 80%.". This value appears to refer to the volumetric ice content of the mineral soil C5 between ice wedges, rather than to an average representative value for a terrain unit or cross-section that includes both the icy soil matrix and ice wedges. At the scale of the modelling, this is what is pertinent, otherwise the contribution to ice content in the upper permafrost from ice wedges is not accounted for.

Authors' reply: For this single-point case for model evaluation, our goal is not to retrieve realistic excess ice melt, but rather to compare the model results from this study and from Westermann et al. (2016). Initializing realistic excess ice condition does not help the model evaluation in this case because the Lena River Delta has observed hardly any surface subsidence yet, making model-observation comparisons inapplicable. Alternatively, we make model-to-model intercomparisons to evaluate our developed physics and sub-grid representation. So we initialize excess ice strictly following that in Westermann et al. (2016). As a result, our sub-grid representation simulates comparable surface subsidences for each sub-grid landunit compared to Westermann et al. (2016), proving the reasonability of our developed sub-grid representation of excess ice. We have added the above clarification in the main text (Line 157-160).

6. Line 106: "The added ice is evenly distributed within each soil layer". In Figure 3, ice it not depicted as evenly distributed in the cross-sectional diagrams. Tile 4 shows large ice wedges, tile 3 a discontinuous (across the landunit) body of ice. The model does not represent ice in this way. These diagrams should reflect that ice is evenly distributed and consistent with the depictions showing "Present" and "Future" conditions.

Authors' reply: Although in the schematic figure and in reality, the ice is not distributed evenly, the framework of CLM and our developed sub-grid representation is able to convert this uneven distribution of excess ice into evenly-distributed excess ice landunits in the CLM. The relative locations of excess ice bodies does not matter because CLM does not include horizontal heat
and water fluxes (we have mentioned it in the discussion section). The set-up of excess ice in
the CLM can be treated as "squeezing" all excess ice (of the same type) into a part of grid point
with evenly-distributed excess ice and the other part of the grid point without excess ice.

**General**

Permafrost soils usually contain large amount of ground ice. Its melting has significant impacts
on infrastructure, landscape and hydrology. Ground ice also affects the timing and speed of
permafrost thaw. This paper modelled the effects of ground ice on permafrost thaw using a sub-
grid representation in the Community Land Model. They first test the implementation in Lena
River delta. It shows that using three land units of different ground ice provides more realistic
results than using one average ice land unit. The modelled thawing depths also very different
among the three land units and from using the average ice content. Then they implemented the
representation across the circum- arctic region using four land units (no ice, low, mid and high
ice) and compared with the results using average ice content. The results shows more realistic
pathways of permafrost degradation and a different total area with permafrost comparing to
using average ice. The circum-arctic excess ice data are rough, the CAPS dataset is a very broad
generalization of the complex ground ice conditions and how to use the dataset is not
straightforward. However, this study does show some progress to include ground ice in a more
realist way than previous studies (no excess ice, or using average for an entire grid) and it
provides a general range of the large-scale impacts of such sub-grid differences. The paper is
well prepared in language and figures.

Authors' reply: Thank you very much for your valuable comments. We agree that the rough
excess ice dataset is the main challenge when we conducted this study. Unfortunately, the
CAPS data is still the best excess ice data available on the global scale although it was released
more than 20 years ago. In this way, we have to design a tiling scheme to fit the CAPS data
into the sub-grid framework we developed, which is not straightforward and contains fairly
empirical estimates on excess ice contents and located depths. Although with the challenges on
the initial condition of excess ice, we manage to convey through this manuscript that a sub-grid
scale modeling of excess ice in the global land models is necessary for retrieving the permafrost
dynamics in the circum-arctic regions, and we have had the modeling tool prepared before the
new generation of excess ice dataset becomes available.

**Major points**

The test study shows very different active-layer thicknesses among the three land units and
from the one-unit with average-ice (Figure 4). The paper did not provide much about the results
of active-layer thickness for the circum-arctic modelling. It would be important to add this part
in the results and analysis. Observations on ground subsidence is sparse and highly depend on
the local conditions. An improved modelling of active-layer thickness would provide some
support evidence about the usefulness of including excess ice in sub-grids.

Authors' reply: The reason we did not mention the difference of active layer depth brought by
the excess ice in the global case is that it is somewhat complicated because of a technical rather
than scientific reason. Theoretically, the presence of excess ice makes the permafrost thermal
regime more stable and a shallower active layer. However, it does not always show in the
modeling case, because the model initializes soil into discrete layers that are with different
thickness. For most land models, the thickness of each soil layer is not the same from top to
bottom. Usually, deeper soil layers are also thicker. In the original soil set-up of the CLM5, the
typical soil layer thickness for the depth between 0.5 to 1 meters is 0.15 meters, while that for
the depth between 3 to 4 meters is more than 0.5 meters. In this way, for the regions with a thicker active layer (e.g. > 2 meters), the presence of excess ice is not associated with a shallower active layer simply because the above soil layer is too thick (which also means the chunk of soil is bigger) to make the stable thermal regime distinguishable. We have now added some discussion in the main text to give readers some more insights.

"Compared to the grid average ice case, even more permafrost areas are sustained in the subgrid ice case" (Line 313-314). However, Figure 9 shows the permafrost area difference between sub-grid case and no ice case is similar to the difference between the average ice case and no ice case before the 2050, after that the latter reached about 1 million km2. That means the permafrost areas under average ice case and sub-grid ice case are similar before the 2050s. After that, the modelled permafrost area under average ice case is larger than under sub-grid ice case. In the last two panels in Figure 7, the shaded area in the second panel seems larger than the second panel. That is not consistent with the results in Figure 9. Not sure whether my understanding is correct. Any way, it would be useful and interesting to provide more explanation and analysis about the differences among these three cases (no ice, average ice and sub grid ice).

Authors' reply: In figure 9, we compared the actual area of permafrost in the sub-grid scale. For example, for a certain grid point with a total area of 0.2 million km2, only a landunit with 20% area weight has permafrost remaining (ALT <6.49 m). Then the area of permafrost for this grid point is 0.02 million km2. But in figure 7 and 8, we compare the permafrost degradation on the grid scale. In figure 7, the complete degradation of permafrost refers to the condition that all the sub-grid landunits in one grid cell are without permafrost. In figure 8, a grid cell is considered "discontinuous permafrost" if some landunit has permafrost while some others not. We have added more content in the figure caption to prevent misunderstandings.

The data about ground ice is rough and how to use the current data is based on some assumptions or artificial choices. It would important to indicate that uncertainties more clearly in the text (the paper already indicated that at different places).

Authors' reply: We have added more discussion on the uncertainty because of the excess ice initialization.

Minor points

Line 28-29: delete "enhance" or "improve".

Authors' reply: We have made the change as you recommended.

Lines 42-44, "The existence of excess ice and its distribution in permafrost can significantly affect the rate of permafrost thawing". It would be useful to add some references here.

Authors' reply: We have added more references.

Line 58: "over generations". It seems strange to say model versions as "generations". It would be clearer to say "in recent years" or so.

Authors' reply: We have made the change as you recommended.

Line 67: "Separate from this", revised to "In addition"

Authors' reply: We have made the change as you recommended.

Line 71-74. Check the grammar for this long sentence.

Authors' reply: We have checked the grammar.

Line 74-95: "the depth distribution of ground ice can vary substantially on the order to 10-50 meters horizontally 75 and 10 meters vertically". Is the depth to the top of ground ice or also including the thickness of ground ice? Probably you want to say both. Check and consider revising the sentence.

Authors' reply: Actually here we just mean the depth of ground ice rather than both the depth and thickness since it is what the cited studies brought.

Line 165: "Satellite Phenology (SP) mode", I do not know what is that. Some explanation would be helpful.

Authors' reply: We have had an explanation for that. SP mode means it does not involve slowly evolving biogeochemical processes such as soil carbon accumulation (Line 180).

Line 220: "Have the same area fraction of low ice landunit", You may add "(20%)" to make it clearer. What is the reason behind this assumption?

Authors' reply: We make this assumption based on the fact that segregated excess ice is distributed widely throughout the permafrost region. So we assume that all the grid points in the CAPS data have some extent of low content ice. Since we define the volumetric content of excess ice in the low ice landunit as 25%, and the lowest category of excess ice in the CAPS data has 5% in volumetric excess ice content, we just assume that this 5% excess ice is contributed by 20% area weight of low content excess ice that is 25% in volumetric excess ice content.

You must have a percentage of land as no excess ice as the total percentage is less than 100% in Table 2 (e.g., for 5% CAPS, the no excess ice area would be 80%). If that is the case, it would be clearer to indicate the no excess ice areal percentage in Table 2, and the scheme actually uses four landunits (as shown in Figure 1) rather than three. For the grid-average ice case, you used the average of the three land units (Line 242) or the four land units?

Authors' reply: We have made the change in Table 2 as you recommended.

Figure 3. The legend is in km2. You may provide the area of a grid or using % of the area of a grid.

Authors' reply: Because the grid cell with a lower latitude has a larger area. We think using km2 can provide more information here.

Line 259-260: "A small amount of excess ice (24kg/m2) melts during the spinup period", which case?

Authors' reply: It is the average ice single-landunit case. We have added such information to the sentence to make it clear.

Lines 302-303: "We define the permafrost degradation in this study as when all the landunits in one grid cell has an active layer thickness greater than 6.5 meters". That is different from the sentence in line 238. Probably the sentence in lines 302-303 is for how you treat the grid in figure 7. If so you can indicate its applications.

Authors' reply: It is a matter of scales. In this study, only global simulation has permafrost degradation condition analyzed. For figure 7 and 8, we addressed analysis on the grid scale, and we regard full permafrost degradation when the permafrost in all landunit in one grid point has disappeared (ALT > 6.5m). For figure 9, we addressed analysis on the landunit scale to compare the actual permafrost area. In this way, we calculate the area of each landunit with permafrost degraded (ALT > 6.5 m). We have reworded these sentences to make this point clear.

Line 350: "as projected until 2100", probably revise to "as we modelled". No observations beyond present.

Authors' reply: We have made the change as you recommended.

Line425, 438: "modelling", "modelled", be consistent with "Modeling" and "Modeled"

Authors' reply: We have made the change as you recommended.

[revised manuscript text omitted]

---

## Author Response (AR2)

Dear editor,

Thank you very much for your valuable review and point-by-point comments. We have revised the manuscript carefully following your comments and suggestions. The revised manuscript and the supplemental material with marked changes are included as follow.

Sincerely,
Lei Cai, on behalf of all coauthors.

[revised manuscript text omitted]
. Note that the subsurface drainage is unlikely to happen in reality since the permafrost at these two sites are much thicker than 10 m. Meanwhile, this has not occurred in excess ice cases yet, making the soil wetter in excess ice cases than in control cases. These responses in permafrost temperature and moisture after permafrost degradation are consistent with the results in Lee et al. (2014), suggesting that the excess ice physics developed in CLM4.5_EXICE performs reasonably in a sub-grid manner in CLM5. Among the three excess ice cases, the "LL" case shows the strongest responses in both soil temperature and soil water content. On the other hand, the effects of excess ice in soil temperature and moisture are weaker in the "ML" and "HL" cases, where the same amount of excess ice is distributed more localized within a fraction of the grid.

Both sites exhibit active layer depth of around 0.5 m by the end of the spinup and active layer thickness does not increase substantially during the historical period (Black lines in Figure S2 and S3). For this reason, excess ice is incorporated one meter below the surface. No excess ice, therefore, melts during either the spin-up or the historical period simulations. Excess ice starts to melt around the 2070s in NSA_LL, while the timing is delayed for about 25 years in the other two cases for the same site (NSA_ML and NSA_HL; Figure S4). It is because the higher content of excess ice covering a smaller area takes longer to absorb enough latent heat of fusion from the atmosphere before it can start melting. Excess ice in NSA completely melts away in the 2170s and the exact timing of which varies slightly (< 5 years) between cases. In Yakutsk, excess ice starts to melt earlier, but with a slower rate compared to NSA. Similar to the NSA cases, Yakutsk_ML and Yakutsk_HL exhibit delays in the timing of excess ice melt compared to Yakutsk_LL. Excess ice in Yakutsk_LL completely melts in the 2170s, while the timings of excess ice melting in Yakutsk_ML and Yakutsk_HL is delayed for about 10 to 15 years, respectively (Figure S4).

Excess ice melting supplies extra water to subsurface water storage, increasing soil water and eventually converting to runoff. The increases in surface runoff correspond well in timing with excess ice melt (Figure S4). Earlier permafrost thaw timing in control cases causes an earlier increase in subsurface runoff and a decrease in the surface runoff than in excess ice cases. On the other hand, when the active layer depth reaches below the deepest soil layer in excess ice cases, more soil water from melt ice leads to the higher subsurface runoff compared to that in control cases. Among the three excess ice cases, the "LL" cases consistently exhibit the strongest and earliest responses in both surface and subsurface runoff as excess ice melts, being consistent with their earlier start of excess ice melt.

[Figure]

**Figure S4: Time series of excess ice (kg  m⁻²), as well as the difference of surface runoff (mm month⁻¹) and subsurface drainage (mm/month) from the three excess ice cases to control cases. A 15-year moving average is applied before plotting both the surface and underground runoff.**

---

## Editor Decision (ED2)

[revised manuscript text omitted]

Present

Future

Surface subsidence

Soil Depth

Tile 1

Tile 2

Tile 3

Tile 4

Bedrock

Tile

Please add descriptions of the brown and green layers to the legend below

R2: Makes a good point here. As the grid cell size increases with latitude, there can be an increase in area simply because the cell is larger, and not because the amount of area is greater with respect to the area of the grid cell. Cells at different latitudes will have different % areas for the same reported area. Please show as % area to normalize. This makes it easy for the readers to "combine" the three maps and get a better sense of the contribution per grid cell of the three ice landunits. This would make it easier for the reader to agree with statements like that on L351-352 regarding the fraction of ice cover within a grid cell.

low content excess ice mid content excess ice

Low ice landunit

Mid ice landunit

Area (km²)

**Figure 3. Schematic representation of the sub-grid excess ice initialization scenario, and maps showing the area occupied by different excess ice landunits, i.e. the initial condition of excess ice in the global simulation.**

[Figure]

**Figure 4. Annual freeze-thaw state for the three terraces for the triple-landunit case, as well as for the average ice single-landunit case.**

[Figure]

**Figure 5. Grid-mean excess ice melt since 1900 for the single-point cases over the Lena river delta**

**with and without the sub-grid excess ice initialization.**

[Figure]

**Figure 6. Maps showing sub-grid surface subsidence (m) in 2000, 2050, 2100 in the low, mid, and high excess ice landunits in the sub-grid ice case.**

[Figure]

**Figure 7. Maps showing the year of completed permafrost degradation (upper set of three maps), as well as the differences between cases (lower set of two maps). The purple color indicates the existence of permafrost in these grid  by 2100. The difference in years is provided only for grid cell with completed permafrost degradation before 2100.**

[Figure]

| No ice case | Grid-average ice case | Sub-grid ice case |

- 🟦 Remaining permafrost   🟪 Partially degraded permafrost, without talik
- 🟥 Partially degraded permafrost, with talik   🟥 Fully degraded permafrost, without year-round unfrozen soil
- 🟥 Fully degraded permafrost, with year-round unfrozen soil

**Figure 8. Maps of different stages of permafrost degradation diagnosed from the model output by**

**the year 2100.**

The colors used for "Remaining permafrost" and "partially degraded permafrost" are too close. Please better differentiate these two classes, and probably the others as well. As per our instructions to authors, you may find ColorBrewer 2.0 is useful for generating a helpful color scheme for these maps.

The text defines what fully degraded permafrost is, but please define somewhere what the difference is between "with" and "without" year-round frozen soil. In particular, as seasonal frost is expected within the entire model domain, how can there be year-round unfrozen soil? Probably best to define the 5 stages within one paragraph so that the reader has a clear understanding of the terminology.

[revised manuscript text omitted]